# Telomerase RNA plays a major role in the completion of the life cycle in *Ustilago maydis* and shares conserved domains with other Ustilaginales

**Juan Antonio Sanpedro-Luna[1], José Juan Jacinto-Vázquez[1], Estela Anastacio-Marcelino[2], Carmen María Posadas-Gutiérrez[1], Iván Olmos-Pineda[3], Jesús Antonio González-Bernal[4], Moisés Carcaño-Montiel[2], Leticia Vega-Alvarado[5], Candelario Vázquez-Cruz[1,2], Patricia Sánchez-Alonso[1,2]***

**1** Instituto de Ciencias, Posgrado en Microbiología, Benemérita Universidad Autónoma de Puebla, Puebla, México, **2** Instituto de Ciencias, Centro de Investigaciones Microbiológicas, Benemérita Universidad Autónoma de Puebla, Puebla, México, **3** Facultad de Ciencias de la Computación, Benemérita Universidad Autónoma de Puebla, Puebla, México, **4** Department of Computer Science and Engineering, The University of Texas Arlington, Arlington, Texas, United States of America, **5** Instituto de Ciencias Aplicadas y Tecnología, Universidad Nacional Autónoma de México, Ciudad Universitaria, Ciudad de México, México, México

\* maria.sanchez@correo.buap.mx

## Abstract

The RNA subunit of telomerase is an essential component whose primary sequence and length are poorly conserved among eukaryotic organisms. The phytopathogen *Ustilago maydis* is a dimorphic fungus of the order Ustilaginales. We analyzed several species of Ustilaginales to computationally identify the TElomere RNA (TER) gene *ter1*. To confirm the identity of the TER gene, we disrupted the gene and characterized telomerase-negative mutants. Similar to catalytic *TERT* mutants, *ter1Δ* mutants exhibit phenotypes of growth delay, telomere shortening and low replicative potential. *ter1*-disrupted mutants were unable to infect maize seedlings in heterozygous crosses and showed defects such as cell cycle arrest and segregation failure. We concluded that *ter1*, which encodes the TER subunit of the telomerase of *U. maydis*, have similar and perhaps more extensive functions than *trt1*.

## Introduction

Telomerase is a ribonucleoprotein complex that synthesizes telomere repeats. It has two core components: the catalytic protein subunit, which has reverse transcriptase activity (*te*lomerase *r*everse *t*ranscriptase; TERT), and the RNA moiety, which is noncovalently joined to TERT and is known as the *te*lomerase *R*NA *c*omponent (TER or TERC) [1,2]. TER contains a sequence that acts as a template for retrotranscription and extends the telomere 3'OH overhanging strand during the late S-phase of the cell cycle [3]. TERT is the limiting factor for telomerase regulation in higher eukaryotes, and many studies of TERT regulation have been performed in the context of the development of multicellular organisms; however, these same

**Data Availability Statement:** Raw sequencing reads and expression are available from the Gene

Expression Omnibus of NCBI (accession number: GSE184051). Nucleotide sequence data reported are available in the Third-Party Annotation Section of the DDBJ/ENA/GenBank databases under the accession number TPA: BK059259 (will only be available after acceptance).

**Funding:** PS-A Proyecto: SAAM-NAT15-I Vicerrectoría de Investigación y de Estudios de Posgrado Benemérita Universidad Autonoma de Puebla http://www.viep.buap.mx/ The funders had no role in study design, data collection and analysis, decision to publish, or preparation of the manuscript. PS-A Proyecto: Oficio No. VIEP/2505/16 Vicerrectoría de Investigación y de Estudios de Posgrado Benemérita Universidad Autonoma de Puebla http://www.viep.buap.mx/ The funders had no role in study design, data collection and analysis, decision to publish, or preparation of the manuscript. JAS-L Beca CONACYT No. 620317 https://conacyt.mx/ Consejo Nacional de Ciencia y Tecnología, Mexico. The funders had no role in study design, data collection and analysis, decision to publish, or preparation of the manuscript. JJJ-V/PS-A Proyecto: 201901042C Laboratorio Nacional de Supercómputo del Sureste de México https://lns.buap.mx/ The funders had no role in study design, data collection and analysis, decision to publish, or preparation of the manuscript.

**Competing interests:** The authors have declared that no competing interests exist.

studies have been performed to a limited extent for TER [4,5]. In higher eukaryotes, TERT is highly regulated but is expressed in variable quantities in tissues that require abundant renovation or high proliferative potential, such as germ and stem cells, and in cancer cell lineages [6–8]. In somatic tissues, which are mainly composed of terminally differentiated cells with low replicative potential, telomerase activity is practically undetectable because TERT is downregulated progressively as development programs advance [7–9]. In lower eukaryotes, telomerase activity is instead observed during each cell cycle for an undefined number of replication cycles because the catalytic subunit is not downregulated [10]. TER, the other telomerase core subunit, is a functional long noncoding RNA that is poorly conserved in length and sequence among eukaryotes; its length varies from 147 nt in *Tetrahymena paravorax* to 2,045 nt in *Mycosphaerella graminicola* [11,12], but its secondary structures, i.e., the template domain, pseudoknot, stem-terminus and TER boundary element, are conserved in all TERs described to date and are essential for holoenzyme assembly and function [13,14].

In all eukaryotes, a lack of any of the telomerase core subunits or holoenzyme downregulation produces telomere shortening after every DNA replication round; when telomeres become too short, replication ceases, and senescence is triggered [7,15,16]. However, in lower eukaryotes, as replication continues, cells enter a crisis state, and cell death occurs. In budding yeast, a small subpopulation of cells that are capable of maintaining telomeric function in the absence of telomerase by recombinational pathways emerges and slowly proliferates [17]. Over time, these survivors develop two different terminal restriction fragment (TRF) patterns, according to which they are designated as Type I and Type II survivors. The chromosome ends in Type I survivors are composed of major amplifications of Y' subtelomeric sequences interposed with small tracts of telomere repeats, whereas Type II survivors have long, heterogeneous tracts of telomeric repeated sequences [18].

Previously, TER has been studied in several different contexts: in mammalian cancer cells, TER has extratelomeric functions in angiogenesis and cancer metastasis [19]; in zebrafish, TER is involved in myelopoiesis; and hTER seems to have regulatory functions on *HIST1*, which encodes histone 1C [20,21]. In yeast, TER/TLC1 interacts with the Ku70/80 heterodimer in a TERT-independent manner and has a modest impact on telomere maintenance and silencing [22]. Additionally, studies of TER have produced insights into the importance of splicing mechanisms for achieving functional TER in Ascomycetes [23]. The structure of TER is an attractive subject of study because of the richness of information it can provide about the ways an essential noncoding RNA can preserve its functionality and activity as a structural and possible regulatory genetic entity over evolutionary time. It has been reported that TER sequence domains and motifs from closely related species are conserved, as in *Saccharomyces sp. "sensu stricto"* by Dandjinou *et al.* [24], *Plasmodium species* by Chakrabarti *et al.* [25], and Pezizomycotina by Qi *et al.* [12]. TER/TLC1 was alleged to be the limiting subunit of telomerase in *Saccharomyces cerevisiae*, as TERT is in higher eukaryotes [9,26], but TER has been little studied in Basidiomycetes, a fungal group that includes dimorphic fungi.

*U. maydis* is a basidiomycetous fungus that is widely used for studies of recombination [27] and phytopathogen-plant interactions [28,29]. It has been used by several research groups as an alternative or complementary model system for studies on telomere maintenance and telomerase metabolism [30–32]. *U. maydis* is advantageous for telomere/telomerase research because it undergoes developmental transitions, existing as spores, yeast-like cells, and mycelia. The ortholog of the TERT gene in *U. maydis*, *trt1*, was characterized in a previous work by Bautista-España *et al.* [32]. Trt1 is responsible for telomere maintenance and genome stabilization in the fungus. *trt1* disruption causes a lack of telomerase activity, telomere attrition, slow growth, poor proliferative potential, crisis, and cell death; however, it also allows the selection of ALT-dependent survivors, both Type I-like and Type II-like, that support telomere function

by different recombinational pathways, as reflected in their TRF patterns. In *U. maydis*, *trt1* disruption causes poor teliospore formation in heterozygous crosses and the essential absence of tumors and teliospores in homozygous crosses. Additionally, it has been reported that *U. maydis* contains a set of telomere-associated proteins that participate in their organization and homeostasis, which have homology with those of other model eukaryotes, suggesting that the chromosome end maintenance mechanisms have similar properties [31,33]. Slight changes in *trt1* expression occurred along the pathogenic transition in *U. maydis*, as well as a clear repression of the *UMAG_03168 locus*; this *locus* harbors the transcription start site of the putative *ter1* gene [34], suggesting that both components might have an important role during cell differentiation independent of telomere lengthening, as occurs in other organisms [20,35–37].

Here we undertake the identification of the *TER* gene in *U. maydis* based on comparative genomics strategies; sequences resembling the template domain of telomerase RNA were used to find syntenic segments conserved in genome sequences of several members of Ustilaginales. The putative gene, here named *ter1*, was on intergenic sequences of a synthenic region conserved in the genome of the Ustilaginales studied here, and all shares the domains essential for TER secondary structure and function. *ter1* is transcriptionally expressed in *U. maydis* sporidia, and its disruption recapitulates the phenotypes previously observed for *trt1*-disrupted mutants [32], i.e., telomere shortening, diminished proliferative potential, delayed growth, senescence phenotype, and the eventual rise of survivors. However, the *ter1Δ* clones were frailer than their counterpart *trt1Δ* mutants, had an increased number of nuclear aberrations in surviving cells, and could not infect maize plants in heterozygous crosses nor accomplish the sexual life cycle as *trt1Δ* mutants do. These results led us to propose that the found gene *ter1* is the TER subunit of the telomerase of *U. maydis*, and its disruption generate survivors through similar mechanisms than their *trt1Δ* counterparts.

## Material and methods

### Strains, plasmids, and media

*U. maydis* strains 521 (*a1b1*) and 518 (*a2b2*) were used. We also used as control the replicative plasmid pCM54, which is a pUC18 derivative that contains UARS1 and an ~4 kb chimeric *hph* gene (bacterial *hph* gene under *hsp70* promoter and terminator sequences that confers hygromycin resistance) which was provided by W.K. Holloman, Cornell Weill Medical College, NY, USA. A 2,039 bp cassette flanked by *Bam*HI containing a second version of the chimeric *hph* gene (bacterial *hph* gene under *hsp70* promoter and short terminator sequences) was excised from the plasmid pCM1007 (also provided by W.K. Holloman) and used for the assembly of the disruption cassette. The plasmid pBluescript KS(-) (Stratagene, La Jolla, CA, USA) was used as the backbone for cloning of the isolated DNA fragments or PCR amplified sequences and to assemble the interruption cassette. *Escherichia coli* strain XL1-blue MRF' (Δ(*mcrA*)183 (*mcrCB-hsdSM-mrr*)173 *recA1 endA1 gyrA96 thi-1 supE44 relA1 lac* [F' *proAB lacIq* ZΔM15 Tn10(tetr)]; Agilent, Santa Clara, CA, USA) and DH5α (F⁻ *endA1 glnV44 thi-1 recA1 relA1 gyrA96 deoR nupG purB20* φ80d*lacZ*ΔM15 Δ(*lacZYA-argF*)U169, hsdR17($r_K^-$ $m_K^+$), λ⁻; Thermo Fisher, Mexico), were used as hosts for plasmids. The restriction enzymes used were from Thermo Fisher Scientific (Waltham, MA, USA). LB media supplemented with ampicillin sodium salt (100 μg/ml; Sigma-Aldrich Co., St. Louis, MO, USA) was used to select *E. coli* transformant strains, and bacterial cultures were grown at 37°C overnight. *U. maydis* strains were grown in YEPS media (1% yeast extract, 1% peptone and 2% sucrose), in potato dextrose broth (PDB) or agar (PDA) from Difco (Becton Dickinson and Company, Sparks, MD, USA), or in complete or minimal medium [38] at 28°C.

## Identification of Ustilaginales *loci* containing TER candidates

The genomic sequences consulted were as follows: *Ustilago maydis* strain 521, accession GCF_000328475.2; *Ustilago bromivora* strain UB2112 accession GCA_900080155.1; *Ustilago cynodontis* strain NBRC 9727; accession GCA_001676635.1; *Ustilago esculenta* strain MMT, accession GCA_000819925.1; *Ustilago hordei* strain Uh4857-4, accession GCA_000286035.1; *Ustilago trichophora* strain RK089, accession GCA_001654535.1; *Ustilago tritici* strain Utri01, accession GCA_002993085.1; *Ustilago vetiveriae* strain RK 075, accession GCA_001735935.1; *Ustilago xerochloae* strain UMa702, accession GCA_001736075.1; *Sporisorium iseilematis-ciliati* strain BRIP 60887 a, accession GCA_001748505.1; *Sporisorium reilianum* strain SRZ2, accession GCA_000230245; *Sporisorium scitamineum* strain SSC39B, accession GCA_001010845.1; *Pseudozyma antarctica* strain T-34, accession GCA_000334475.1; *Moesziomyces aphidis* strain DSM 70725, accession GCA_000517465.1; *Kalmanozyma brasiliensis* strain GHG001, accession GCF_000497045.1; *Pseudozyma flocculosa* strain PF-1, accession GCA_000417875.1; *Pseudozyma hubeiensis* strain NBRC 105055, accession GCA_001736105.1; *Pseudozyma tsukubaensis* strain NBRC 1940, accession GCA_001736125.1; *Pseudozyma* sp. F8B2, accession GCA_003004685.1; *Moesziomyces* sp. F5C1, accession GCA_003004715.1; *Moesziomyces* sp. F16C1, accession GCA_003004725.1; and *Melanopsichium pennsylvanicum* strain 4, accession ERS346485. Candidate *loci* encoding the RNA component of the telomerase gene (here named *ter1*) were identified through a search of the entire *U. maydis* genome (GCF_000328475.2) according to the method described by Chakrabarti *et al.* [25]. Briefly, the entire genome sequence of *U. maydis* was examined using both the Motif Finding Automaton (MFA) from Pérez *et al.* [39] and SnapGene® v.3.2.1 software (from Insightful Science; available at snapgene.com) to find all regions containing at least one and a half copies of the template sequence (5'-CCCTAA-3') in any possible permutation. Sequences identified as telomeres were discarded, and sequences identified as ORFs, structural RNAs and nonconserved sequences; target motifs along with 300 bp of flanking sequences on each side were recovered and saved in a local database set. Each sequence was then aligned to all genomes of the Ustilaginales taxon group available on April 30, 2019, using blastn option of BLAST [40] software of NCBI. The highest-scoring single *loci* conserved in all organisms were sought, and 21 candidates were obtained. Using SnapGene® v.3.2.1 and the genome annotation of *U. maydis* https://ftp.ncbi.nlm.nih.gov/genomes/all/GCF/000/328/475/GCF_000328475.2_Umaydis521_2.0/GCF_000328475.2_Umaydis521_2.0_genomic.gbff.gz as a reference, the 21 candidate *loci* were re-examined both to predict the ORFs flanking the intergenic regions and to perform a synteny analysis of nearly 15 kb on both sides of the putative template domain in the probed genomes, even including putative ORFs without annotation. To achieve this aim, approximately 30 kb of each genome of the chosen organisms was cropped, and the cropped sequences were aligned using blastn. After the analysis, the complete intergenic sequences harboring the TERs were recovered, and their lengths were determined (ranging from 784 to 2,669 nucleotides); then, multiple sequence alignment was performed using the online version of MAFFT software (https://mafft.cbrc.jp/alignment/server/index.html) [41] to find conserved sequence motifs. A perfect match of the putative template domain was observed within the putative *loci ter1*, nucleotide sequence data reported are available in the Third-Party Annotation Section of the DDBJ/ENA/GenBank databases under the accession number TPA: BK059259.

## Transcriptional expression

Before *ter1* gene disruption, gene expression data were sought from the NCBI database, using the *ter1* sequence as a query. To confirm the existence of those mature transcripts derived

from either the 3' of the hypothetical gene *UMAG_03168* located upstream of the *ter1 locus* or the 5' ter1 transcript, a series of primers was designed to perform the reverse transcription assays (S3 Table), and Oligo $dT_{18}$ (Thermo Fisher Scientific) was also used. 1 μg of RNA previously extracted as described below, added with 100 μM reverse oligo, and lead to a final volume of 12 μl, was incubated at 65˚C for 5 min. The previous mixture was added with 4 μl of 5X Buffer (RevertAid H Minus First Strand cDNA Synthesis Kit, Thermo Fisher Scientific), 20 U of RiboLock RNAse Inhibitor (Thermo Fisher Scientific), 10 mM dNTPs, and 200 U of RevertAid H Minus Reverse Transcriptase (RevertAid H Minus First Strand cDNA Synthesis Kit), the reaction was incubated at 25˚C for 5 minutes, followed by incubation at 45˚C for 60 minutes and finally at 70˚C for 10 minutes. 3 μl of cDNA from the previous reaction or 300 ng of DNA used as a positive control were added with 16 μM forward primer, 16 μM reverse primer. PCR reactions were performed using Platinum® *Pfx* DNA Polymerase (Thermo Fisher Scientific) according to the manufacturer's instructions.

## Nuclei extraction

For the extraction of nuclear RNA ($RNA_N$) and total RNA ($RNA_T$), nuclei were obtained based on Groner and Phillips [42]. Briefly, 100 μl of an optimal preinoculum of *U. maydis* strain 521 was added to 5 ml of YEPS broth and incubated at 30˚C for 12 h with aeration (200 rpm) to obtain the inoculum, from which 100 μl was taken and added to 100 ml of YEPS and incubated as above until $OD_{600}$ = 0.8. Cells were centrifuged at 3,500 rpm/10 min at room temperature. The supernatant was discarded, and the pellet was suspended in 100 ml of prewarmed YEPS broth and incubated at 30˚C for 4 h under constant shaking (200 rpm) to obtain exponentially grown cells. The culture was centrifuged as above, and the pellet was recovered and suspended in 5 ml of SCS solution (50 mM sodium citrate, 1 M sorbitol, pH 5.8) and washed twice. For each wash, centrifugation was performed for 7 min at 2,000 rpm, and after the last wash, the pellet was resuspended in 2 ml of SCS, and 50 mg/ml of lysing enzymes from *Trichoderma harzianum* (L1412, Sigma-Aldrich Co.) were added. After 20 min of incubation at 37˚C, protoplasts were centrifuged at 1,000 rpm/10 min, recovered, and washed twice with 20 ml of 1 M sorbitol at 4˚C. After the last centrifugation, the pellet was recovered, 2 ml of NIB buffer (17% glycerol, 50 mM MOPS pH 7.5, 150 mM potassium acetate, 2 mM $MgCl_2$, 500 μM spermidine and 150 μM spermine) was added, and the pellet was left on ice for 30 min. For centrifugation, 3 ml of sucrose cushion (2 M sucrose and 2.5 μl of 1 M $MgCl_2$) was placed in a tube for each 2 ml of lysate and centrifuged at 10,000 rpm for 15 min. The supernatant was removed and stored at -80˚C for other purposes. The pellet was suspended in 3 ml of SDS buffer (0.1 M NaCl, 0.02 M EDTA, 0.02 M Tris-HCl pH 7 and 0.5% SDS) and homogenized, and subsequent steps were performed as indicated by the RNeasy Mini Kit (Qiagen). The samples were evaluated by gel electrophoresis, and the optical density of each sample was read to analyze the sample integrity. Samples were stored at -80˚C.

## Transcriptome analysis

Analysis of the regions comprising *UMAG_03168*, *ter1* and *UMAG_03169* from strain 521 of *U. maydis* was performed. Briefly, poly(A+) RNA and total (ribo−) RNA were obtained from the nuclear fraction of the wild-type 521 strain of *U. maydis*. cDNA libraries were prepared using the TruSeq Stranded mRNA Library Preparation kit and Truseq Stranded Total RNA Gold Prep Kit (Illumina Inc.), respectively, and sequenced by DNAlink using the Illumina NovaSeq 6000 platform, generating paired-end reads of 101 bp. The quality control of reads was carried out using FastQC v0.11.9 [43] subsequently, the adapters and low quality reads were eliminated using Trimmomatic v0.36 [44] software with the following parameters: PE,

-phred33, ILLUMINACLIP:$TRIMMOMATIC_DIR/adapters/TruSeq3-PE-2.fa:2:30:10:1:true SLIDINGWINDOW:4:15 LEADING:3 TRAILING:3 AVGQUAL:30 MINLEN:25. Then, rRNA and mitochondrial transcript reads were removed using SortMeRNA v4.2.0 [45] software with default settings, using the rRNA and mitochondrial genome sequences of *U. maydis*. *De novo* transcriptome assembly was conducted with Trinity software v2.11.0 [46] with the following settings:—seqType fq,—min_kmer_cov = 2,—min_contig_length = 100—jaccard_clip —SS_lib_type RF. Read alignment and abundance quantification were performed using Bowtie2 v2.4.2 [47] and RSEM v1.3.3 [48], respectively, using the align_and_estimate_abundance. pl script from the Trinity pipeline with the following settings:—seqType fq,—est_method RSEM,—aln_method bowtie2. Data visualization was carried out with the Integrative Genomics Viewer [49]. Finally, for the analysis of differential expression, the online platform IDEA-MEX (Integrated Differential Expression Analysis MultiExperiment, http://www.uusmb. unam.mx/ideamex [50] was used with the following methods: DESeq2 [51], NOISeq [52], Limma-Voom [53], and EdgeR [54]. Transcripts were considered differentially expressed only if they had a log2FC $\geq$ 2 and an FDR $\leq$ 0.01. Raw sequencing reads and expression data were submitted to Gene Expression Omnibus of NCBI (accession number: GSE184051).

## Nucleic acids extraction and manipulation

DNA extraction from *U. maydis* cells was achieved as follows: 50 ml of YEPS broth was inoculated with the chosen fresh strains. The cell cultures were grown overnight to an $OD_{600}$ = 1.0– 1.2 at 28˚C with aeration; cells were harvested in a 5810 Eppendorf centrifuge with an A-4-62 rotor (Eppendorf North America, Inc., Hauppage, NY, USA) at 2,000 rpm. The culture medium was discarded, and the pellet was frozen and ground in liquid nitrogen with a mortar and pestle. Powder was poured into a 50 ml tube containing 5 ml of lysis buffer (200 mM Tris-HCl, pH 8.5; 250 mM NaCl; 25 mM EDTA; and 0.5% SDS) and mixed thoroughly. DNA was extracted by a phenol-chloroform procedure, conducted in two rounds with 15-minute centrifugation intervals. The supernatant was washed once with chloroform and precipitated with 0.6 v/v isopropanol. The DNA was pelleted by centrifugation at 12,000 rpm in a microfuge; the aqueous phase was discarded, and the pellet was suspended in 100 μl of sterile distilled water and re-extracted with phenol once again. After centrifugation, the DNA pellet was suspended in 50 μl of sterile distilled water. For Southern blot assays, the DNA was quantified by spectrophotometry using a GENESYS 10S UV-Vis (Thermo Fisher Scientific) spectrophotometer according to the manufacturer's instructions prior to and after restriction enzyme digestion. Approximately equal amounts of DNA were used for each lane in the TRF assays.

RNA was extracted as follows: fresh colonies were inoculated into tubes containing 5 ml of YEPS medium and incubated at 28˚C overnight. When necessary, hygromycin (200 μg/ml, (Sigma-Aldrich Co.) was added to the medium; 50 μl of the preculture was added to 5 ml of YEPS medium and incubated for 10 h. Then, an aliquot of 0.3 to 0.5 ml of strain was inoculated into a flask containing 200 ml of YEPS medium (with or without hygromycin for selection) and incubated until it reached $OD_{600}$ = 0.6–0.8. The cultures were centrifuged at 2,000 rpm for 10 min, the supernatant was discarded, and the pellet was frozen in liquid nitrogen and ground to powder with a mortar and pestle. The powder was recovered, and 2 ml of lysis buffer R (0.02 M sodium acetate, 0.5% SDS, and 1 mM EDTA, pH 5.5) was added and mixed thoroughly with the powder by vortexing. Samples were then partitioned into 500 μl aliquots in microtubes. Each sample was mixed with 500 μl of acid phenol (phenol equilibrated with 0.02 M sodium acetate pH 5.5) and placed at 65˚C for 5 min mixing occasionally. The samples were cooled and centrifuged at 10,000 rpm for 10 min at 4˚C, and the supernatant was transferred to another tube. Then, 500 μl of acid phenol:chloroform:iso-amyl alcohol (125:24:1) was

added. The mixture was homogenized and centrifuged at 10,000 rpm for 10 min at 4˚C. The aqueous phase was washed with 500 μl chloroform twice and centrifuged at 10,000 rpm for 5 min at 4˚C; then, 7.5 M ammonium acetate and 2 volumes of absolute ethanol were added, and the sample was mixed vigorously and left at -70˚C for 1 h. RNA was recovered by centrifugation at 10,000 rpm for 10 min at 4˚C. The pellet was washed twice with 70% cold ethanol at -20˚C. Finally, the isolated RNA was dissolved in 100 μl of sterile water. Samples were treated with RNase-free DNase I (Merck KGaA, Darmstadt, Germany) according to the manufacturer recommendations.

## Disruption cassette

Approximately 1 kb of the DNA sequences flanking the putative template domain in the *ter1 locus* were amplified by PCR and cloned in pBluescript KS(-), oligonucleotide primer pairs (Integrated DNA Technologies Inc., San Diego CA, USA) were Cr8S1-5Upp (genomic location, 124,977–125,009) and Cr8S1-5Low (126,126–126,157), which amplify 1,181 bp of the hanging 5' side of the disruption cassette. The amplified DNA fragments contained *Not*I and *Bam*HI cut sites on their 5' and 3' sides, respectively. For the 3' hanging DNA fragment of the cassette, Cr8S1-3Upp (126,791–126,819) and Cr8S1-3Low (127,937–127,965) were used to amplify a 1,175 bp DNA fragment downstream of the putative template domain and bordered by *Bam*HI and *Not*I cut sites. The two fragments were cloned into pBluescript KS(-) to generate plasmids pTer5 and pTer3, respectively. A 1,170 bp DNA fragment was excised from the pTer5 plasmid by *Not*I/*Bam*HI digestion and was subcloned into pTer3 in the *Not*I/*Bam*HI sites. The resulting plasmid was pTerSub5, which contained a *Bam*HI site between the upstream and downstream fragments. Finally, the 2,039 bp DNA *Bam*H1 fragment from pCM1007 containing the *hph* chimeric gene was subcloned into the *Bam*HI site of pTerSub5 to obtain the 4,373 bp disruption cassette in the plasmid pTer1, which was excised from the plasmid by complete *Not*I digestion. The excised fragment was resolved by electrophoresis on an 0.8% agarose gel and recovered from the agarose by the freeze–squeeze method.

## Transformation and confirmation of *ter1Δ* mutants

Transformation was performed as described in Bautista-España *et al*. [32] following the Tsukuda *et al*. [55] protocol, using the 518 (*a2b2*) strain, the linearized disruption cassette, and pCM54 as a positive control. The bottom layer of the plates of selective medium was supplemented with 400 μg/ml hygromycin such that the medium reached a final concentration of 200 μg/ml after the top layer without hygromycin was added. Transformants appeared after five days of incubation on selective medium at 28˚C, and individual clones were streaked on plates of PDA medium without selective pressure and then grown on YEPS plates and in YEPS broth, both containing hygromycin at 200 μg/ml. These cultures were used for observations of telomere shortening after a passage series, for cryopreservation of early-stage transformants in glycerol (50% v/v, -70˚C) and for total DNA extraction as described above. *ter1* disruption in transformants was confirmed by PCR using the oligonucleotide primer pairs Template-Upp (genomic location 126,268–126,287 [irp 188–207 nt]) and Template-Low (genomic location 126,492–126,511 [irp 412–431 nt]) to check for the absence of the template domain. Its replacement by the *hph* cassette was confirmed by PCR using the primer pair 5H-Upp (genomic location 124,812–124,835) and 5H-Low (position on the selection marker 396–419) and the primer pair 3H-Upp (position on the selection marker 1,768–1,788) and 3H-Low (genomic location 129,503–129,524) to amplify DNA fragments spanning from the 5' and 3' ends of the selective marker to DNA regions outside the homologous regions used for the disruption cassette assembly.

## Growth kinetics

Individual colonies of clones and the wild-type 518 strain were inoculated into tubes containing 5 ml YEPS broth and grown at 28°C with aeration to $OD_{600}$ = 1; 1 ml from these precultures was inoculated into 100 ml of YEPS medium and incubated at 28°C with aeration (200 rpm). A series of 1 ml samples were taken every 2 h for 18 h, and the $OD_{600}$ was determined and recorded. Serial dilutions were made to determine the CFU at each time point, plated in YEPS medium, and incubated at 28°C for 24 h. The data were graphed using GraphPad Prism v. 8.4.3. Linear regressions of the log10 CFU vs. time were performed, and the exponential phase data were used to calculate the generation time of the wild-type strain and *ter1Δ* mutants.

## TRF analysis in *U. maydis*

TRF analysis was carried out as previously described by Bautista-España *et al.* [32]. Four micrograms of total DNA from each strain or clone was digested to completion with *Pst*I (Thermo Fisher Scientific), and 500 ng of the digested DNA per lane was electrophoresed in a 1.0% agarose gel and transferred to a nylon membrane (Hybond-N, Invitrogen, Carlsbad, CA, USA) as described by Sambrook *et al.* [56]. Hybridization was carried out in Church and Gilbert [57] solution (0.5 M $Na_2HPO_4$ pH 7.2, 7% SDS, 1 mM EDTA and 1% BSA; Sigma-Aldrich Co.). The probe used in this work was the 232 bp *Hin*CII/*Xho*I DNA fragment from the pUT2 plasmid [30] (accession X77242.1) containing 37 copies of the telomere repeat TTAGGG and 9 bp of the TAS sequence, labeled with digoxigenin-11-uridine-5'-triphosphate (Roche Diagnostics GmbH, Mannheim, Germany, provided by Sigma-Aldrich Co.). Hybridization was performed at 59°C, followed by washes with 0.2X SSPE and 0.1% SDS at 59°C. TRF length was estimated using the ImageJ 1.46r program (Wayne Rasband, NIH, USA).

## Microscopy

*U. maydis* cells freshly cultured in YEPS medium at $2.0 \times 10^7$ cells/ml were harvested by centrifugation at 2,500 rpm for 5 min at 30°C. The medium was discarded, and the cells were washed twice in 200 μl of TBS (50 mM Tris HCl pH 7.5, 150 mM NaCl), with centrifugation for 5 min at 2,500 rpm between washes. The cells were suspended in 100 μl of TBS and mixed thoroughly with 100 μl of fixative solution (50% v/v ethanol, 10% v/v $CH_3COOH$). The suspension was kept at RT for 5 min and centrifuged again to remove the fixative solution. Cells were washed twice with 200 μl of TBS, and the pellet was suspended in 100 μl of the same solution supplemented with 0.2 μg/ml RNase A from bovine pancreas (Sigma-Aldrich Co.). The cell suspension was kept at 4°C overnight, moved to 37°C for 1 h, centrifuged and washed once in 200 μl of TBS. The cell pellet was then suspended in 50 μl TBS with propidium iodide (2 μg/ml; Sigma-Aldrich Co.) and allow to absorb the dye for 6–8 h at 4°C. Stained cells were washed twice with 100 μl of TBS and resuspended in 50 μl of TBS. Then, calcofluor (Sigma-Aldrich Co.) was added at 25 ng/ml TBS, and after 5 minutes of treatment, the excess dye was removed by centrifugation and suspended in 25 μl of buffer for observation under a microscope.

## Plant infection assays

Infection assays in five days-old maize-seedlings line MCS-02 were carried out according to the method previously described by Bautista-España [32], using mixtures of compatible strains wild-type and ter1⁻ to achieve the heterozygous cross *in planta*.

## PFGE

Fresh inoculum of *U. maydis* strains (50 μl) was inoculated into 20 ml of YEPS and incubated overnight at 20–30°C with aeration. When the culture reached $OD_{600} = 1.0$, cells were collected by centrifugation at 2,000 rpm for 7 min, the supernatant was discarded, and the cells were washed twice in 3 ml SCS buffer (20 mM sodium citrate pH 5.8, 1 M sorbitol) and finally suspended in 500 μl of SCS buffer. Then, 100 μl of lysing enzymes from *Trichoderma harzianum* (L1412, Sigma-Aldrich Co.) at 50 μg/ml in CFS were added and allowed to stand for 15 min at 28–30°C without agitation. Protoplasts were centrifuged again for 7 min at 1,000 rpm and resuspended in 500 μl of CFS; then, the suspension was added to 800 μl of 1% low melting point agarose (UltraPure Low Melting Point agarose; Thermo Fisher Scientific) that had been dissolved in 125 mM EDTA, melted, and cooled to ∼37°C; the mixture was homogenized, poured into molds, and allowed to gel on ice for 10 min before being released into 50 ml sterile conical tubes, covered with LET buffer (0.5 M EDTA, 10 mM Tris pH 7.0, 5 μl β2-mercaptoethanol), and incubated at 30°C for 12 h. After this time, the LET was replaced with approximately 4 ml of NDS (0.15 M EDTA, 0.01 M Tris-HCl pH 8.5, 1% N-lauryl sarcosine pH 9.5, 2 mg/ml proteinase K (Sigma P-8044, Sigma Chemical Co.) and incubated overnight at 50°C; then, the NDS was discarded, and the plugs were washed in 50 mM EDTA solution and stored at 4°C until use. For the PFGE run, the plugs were rinsed in 100 mM EDTA solution, placed in the wells of a 1% agarose ultrapure (chromosomal-grade) gel, melted in 0.35X TBE and gelled in the PFGE mold. The wells were covered with LMP agarose melted in 125 mM EDTA. Pulsed field electrophoresis was performed in a Bio-Rad CHEF DRII unit with the following conditions: running buffer TBE 0.35 X, temperature 14°C, agarose gel 1% in TBE 0.35X. Block 1: Pulse 160 s, 2.5 V/cm, 16 h; block 2: Pulse 200 s, 3.5 V/cm, 16 h the running conditions varied as described in the images; running buffer TBE 0.35 X.

## Results

### Computational search for the TER gene in *U. maydis*

Candidate *loci* harboring one and half copies of the 5'-CCCTAA-3' motif in a nine-nucleotide window were identified and compiled, yielding 248 regions in the *U. maydis* genome, which we then searched for TER subunits. After BLAST searching these candidates against all genome sequences from Ustilaginales species, two candidates sharing a putative template domain were recovered. These candidates occupied orthologous genomic positions and had the highest scores. The first candidate matched sequences of the 18S ribosomal RNA gene, which is highly conserved among the Basidiomycota; therefore, it was discarded. The second candidate sequence consisted of 1,626 bp located on chromosome 8 (position 126,081–127,706 nt) that matched an intergenic region (ir) containing the putative template domain that was found toward the 5' ends of intergenic regions among Ustilaginales species. Intergenic regions from these 22 fungal species, comprising the syntenic regions around putative TERs, were selected (Fig 1); our analysis revealed that the additional ORFs and intergenic sequences in all those syntenic regions were matched, which also coincided with the genome annotations of the sequences in the *U. maydis* strain 521 data bank. After manual inspection of putative TER subunits from 21 additional species of Ustilaginales, the potential template domain that once transcribed would be 5' UAACCCUAA 3' was identified (Fig 2).

Once we drew up the boundaries of the intergenic region, multiple sequence alignment of the genome orthologs was performed using MAFFT software. The putative template domain of *U. maydis* lies in a position between 385–393 nt from the intergenic ncRNA gene, here named *ter1*. These domains perfectly matched all putative TERs of Ustilaginales included in

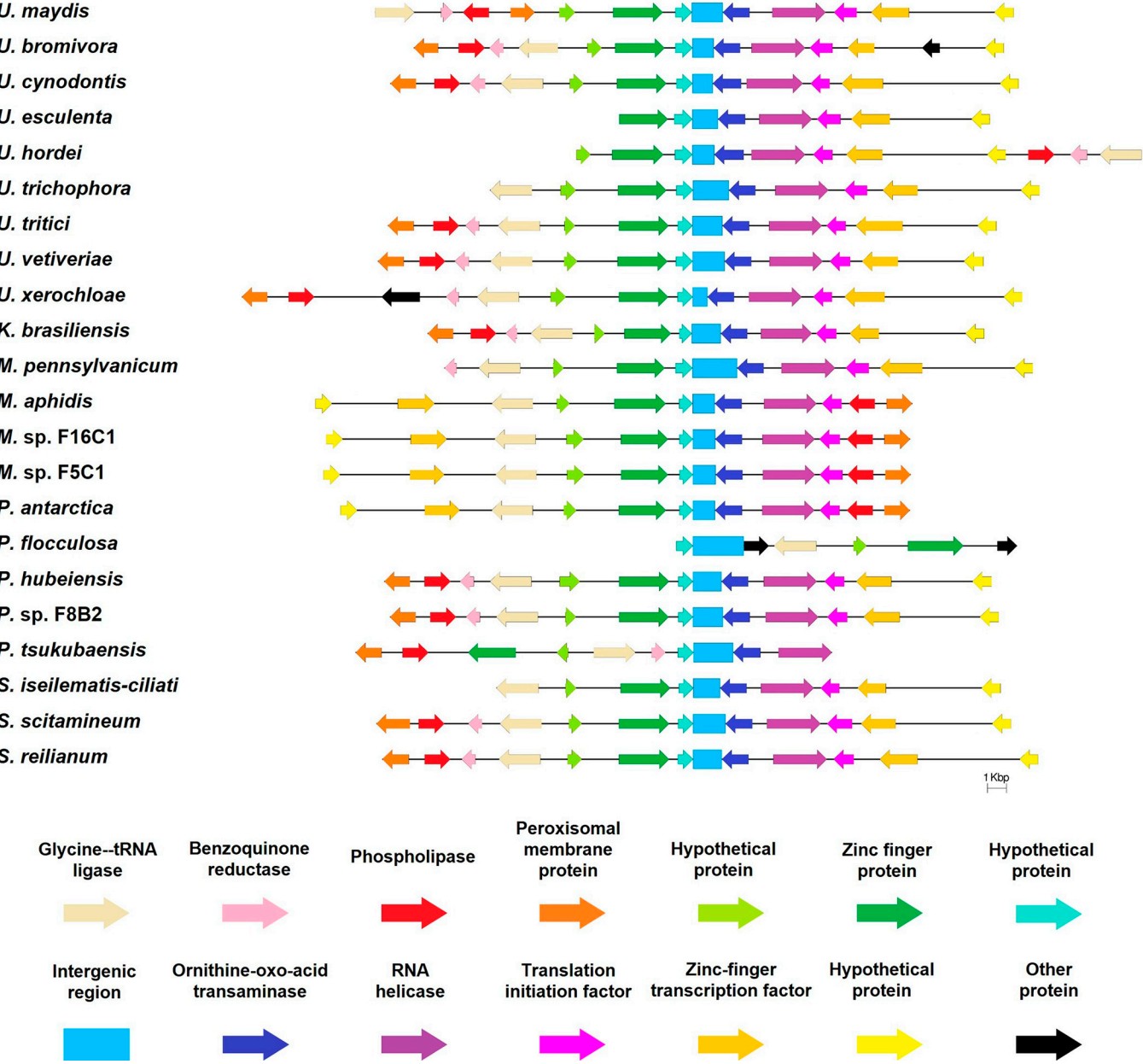

**Fig 1. Synteny map around the *ter1* locus in Ustilaginales species.** Representation of the genes that encode proteins that are located in a widely preserved region around the intergenic area where the *ter1* gene is located (indicated by a blue rectangle). The region shows high conservation, where the same two ORFs flank *ter1* in every examined organism (except of the 3' ORF in *P. flocculosa*).

this study. Other well-conserved elements (CE) at the end 5' of the putative TER of Ustilaginales were CE1 (`TTCGATCY`/`RGATCGAA`), which, along with CE2 (`CGCCCGCC`/`GGCGGGCG`) and CE3, conserve slight variants of `CCTTGACT`/`AGTCAAGG` in its sequence, the domains comprise three sequence stretches of unknown function within of the TER subunit (Fig 2). Downstream of *ter1*, a less conserved T+A-rich DNA sequence spanning 730–780 harbored a pseudoknot domain, which was found in all the species analyzed (S1C Fig). A brief visual inspection allowed us to identify a preserved region (S2A Fig) that resembles the `A`A`GAG`TTGGG`CTCT`G sequence, which creates the P6.1 helix of the CR4-CR5 domain [13,58]

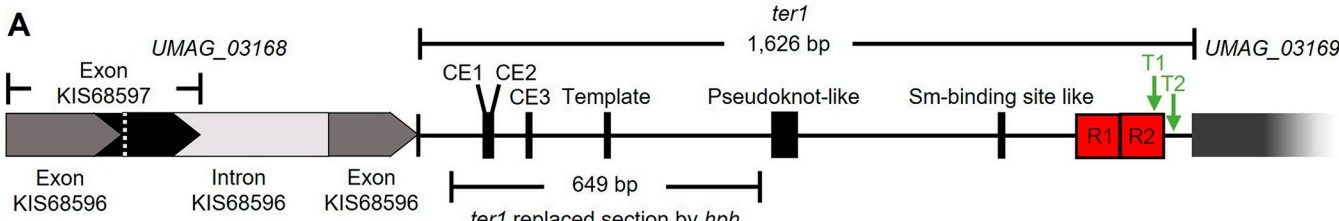

**Fig 2. Preserved elements identified within the *ter1* locus.** (A) Representation of the chromosomal context of *ter1*. The image shows the region corresponding to both isoforms of the *UMAG_03168 locus*, the deleted region and the location of preserved motifs that include essential elements for TER activity obtained from multiple sequence alignment. (B) Multiple alignments of preserved motifs within orthologous intergenic regions of *ter1*. The intergenic regions contain characteristic TER sequences. The sequence of the putative template domain is conserved in all examined organisms. At the 5' end of the region, three preserved elements were identified (CE1–3). Toward the 3' end of the region, the consensus sequence AAU$_{5-6}$CY was identified, which resembles the Sm-binding site of TLC1 in *S. cerevisiae* and thus could correspond to the end of the mature *ter1* transcript.

that is essential for catalytic activity [59,60]. According to the RNAalifold [61] program, in addition, this sequence has the potential to form a stem-loop P6.1-like structure (S2B Fig) and contains some residues within uracil and guanine loops, in the same way as the P6.1 of vertebrates and filamentous fungi [12]. Toward the 3' end, a consensus transcribed sequence AAU$_{5-6}$CY that resembles the Sm (AAU$_5$GG) site of TLC1 in *S. cerevisiae* [24] was found in *ter1* at position 1,208–1,218 (Fig 2). Two repeats of 90 bp in length, here named TRR1 (Telomerase RNA Repeat 1) and TRR2 (Telomerase RNA Repeat 2), that share 96.67% identity were found in *ter1*; their positions spanned from 1,380 nt to 1,469 nt and from 1,472 nt to 1,561 nt and were exclusive to the *U. maydis* TER sequence (Fig 2A). Copies of the polyadenylation sequence signal 5'-AGTAAA-3' reported by Doyle *et al.* [62] were found in the TRR2 repeat embedded in positions 1,537–1,542 and downstream of TRR2 at positions 1,578–1,583 of the *ter1* region (T1 and T2). Because of the findings presented below, we consider the second site to be the main polyadenylation site for the *ter1* TER gene (Figs 3A and S3C). This site is moderately to poorly conserved in the intergenic region of most Ustilaginales analyzed here, a feature previously documented by Ji *et al.* [63] for intergenic regions of yeasts and plant genomes. The search for transcription factor binding sites of *U. maydis* previously described by Basse and Farfsing [64] yielded no target sites nor constitutive splicing consensus motifs were found in the entire *ter1* gene.

## Characterization of the *ter1* transcript

To confirm its identity as a gene, the transcriptional expression of *ter1* needed to be corroborated. First, we examined the dbEST database of NCBI and the transcriptome of the WT strain previously reported [65,66]. To further characterize *ter1*, we analyzed the transcriptomes of both mRNA nuclear transcripts and nuclear RNA-ribodepleted transcripts of strain 521, which were assembled as described previously. From the reads of both nuclear fractions, we found four reconstructed transcripts of the *ter1 locus*, which represent four isoforms containing the structural domains of the TERs. All four isoforms of *ter1* start 122 bp upstream of the start codon of the uncharacterized gene *UMAG_03168* (https://fungi.ensembl.org/Ustilago_maydis/), which has a length of 859 bp and is located on chr 8 (spanning 125,222–126,080 nt); the gene encodes two protein isoforms, 134 aa (KIS68597, accession XP_011389625.1) and 142 aa (KIS68596, accession XP_011389624.1), respectively, which differ according to the splicing of the two exons of 241 and 188 bp and one intron of 430 bp composing the uncharacterized gene (Fig 2A).

The first isoform of *ter1*, here named ter1-i1, is a transcript of 2,475 nt that spans from 125,100 to 127,574; the start site position is consistent for all isoforms reported here (Fig 3B). In agreement with the transcriptome analysis, ter1-i1 is a nonspliced continuous transcript whose abundance is similar in the polyadenylated RNA fraction and the total RNA fraction (Table 1). The second isoform, ter1-i2, is a transcript of 2,347 nt produced from two spliced exons, the first located from 125,100 to 125,764, and the second located from 125,893 to 127,574 (Fig 3B). The reconstructed transcript suggests the use of the noncanonical alternative donor site (D2) TTTCCCC<u>CAAT</u>gtgagcacca (motif conserved among Ustilaginales is underlined, position 125,761–125,770) for splicing (Figs 3A and S3A); the branch (B) site

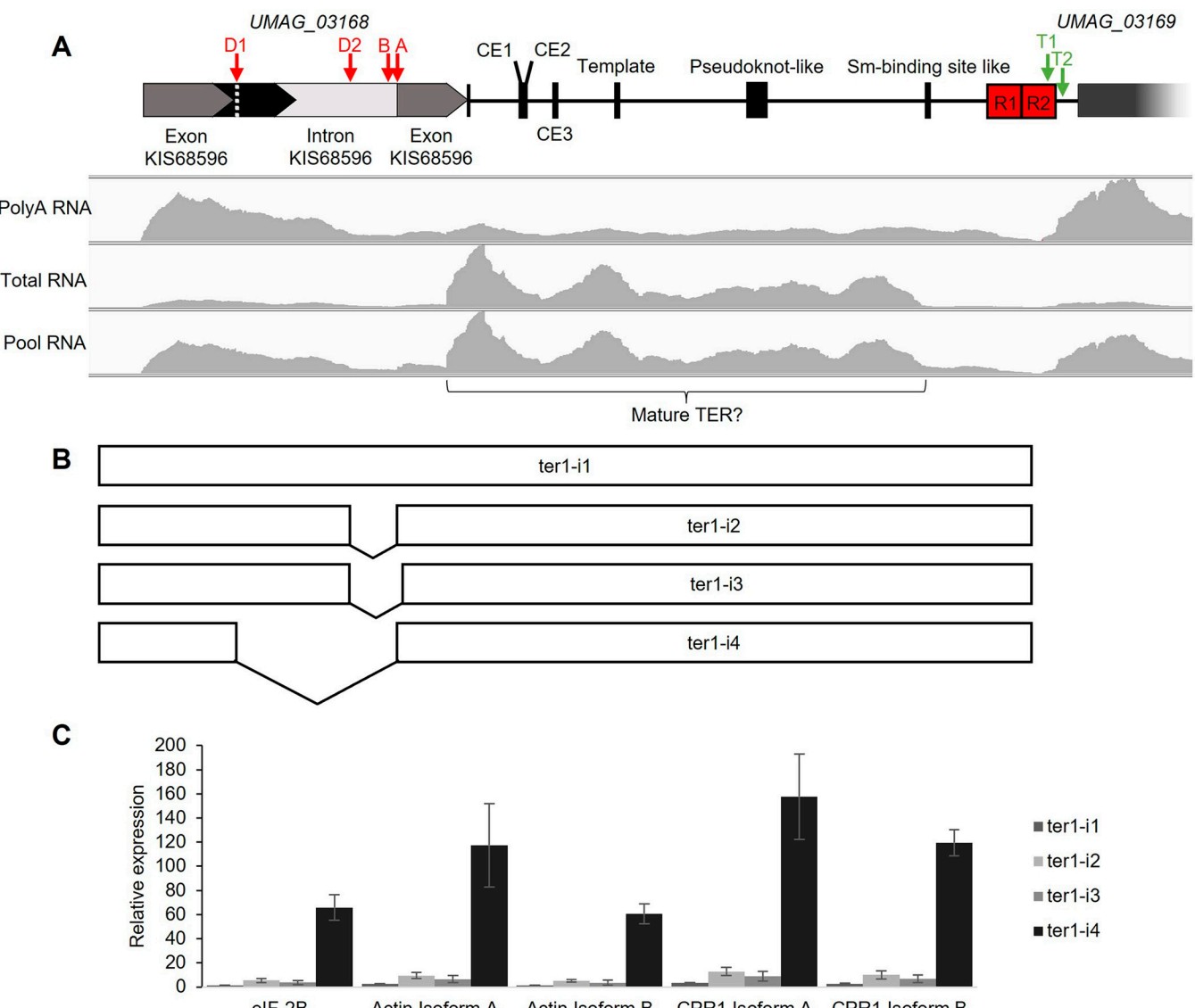

**Fig 3. RNA-seq data analysis of the *ter1 locus*.** (A) Genomic context and conserved elements (upper panel) and density of RNA-seq reads mapped to the *ter1 locus* (lower panel). (B) Transcripts of the *ter1 locus* originating upstream of *UMAG_03168* (represented by white rectangles) reconstructed in this work. (C) Analysis of the relative expression of the isoforms of *ter1*. Quantification of normalized counts was carried out with the EdgeR method for each isoform of *ter1* and of isoforms of housekeeping genes.

**Table 1. Differential expression analysis of *ter1* isoforms.**

| | PolyA RNA vs. Total RNA | | | | | | | |
|---|---|---|---|---|---|---|---|---|
| **Transcript** | **DESeq2** | **padj** | **EdgeR** | **FDR** | **NOISeq** | **prob** | **Limma-Voom** | **padj** |
| **ter1-i1** | 1.11 | 2.04E-17 | 1.10 | 3.63E-14 | 1.08 | 1.00E+00 | 1.10 | 5.61E-04 |
| **ter1-i2** | 2.99 | 6.29E-93 | 2.99 | 1.11E-70 | 2.96 | 1.00E+00 | 3.00 | 1.64E-05 |
| **ter1-i3** | 2.45 | 9.42E-11 | 2.45 | 1.75E-13 | 2.42 | 1.00E+00 | 2.31 | 1.05E-03 |
| **ter1-i4** | 6.62 | 0.00E+00 | 6.62 | 0.00E+00 | 6.59 | 1.00E+00 | 6.61 | 5.09E-07 |

AGCTCACCG (125,863–125,871) and acceptor site AG (125,891–125,892) are the same as those used for the splicing of the KIS68596 transcript. The third isoform reconstructed, ter1-i3, is a 2,332 nt transcript produced by the splicing of two exons; the first exon (664 nt, encoded by 125,100–125,764) and the second exon (1,666 nt, 125,908–127,574). This isoform seems to be the product of alternate splicing using the same donor and branching sites as in ter1-i2 with an alternative acceptor site located 13 nt downstream of the ter1-i2 acceptor site. Similar to the ter1-i2 isoform, its abundance is increased in the total fraction relative to the polyadenylated fraction, with logFC values approximately 3 and 2.4, respectively (Table 1). Finally, the fourth reconstructed transcript, ter1-i4, has a length of 2,045 nt, composed of two exons of 363 bp (125,100–125,462) and 1,682 bp (125,893–127,574) and one intron of 430 bp, making this the largest intron of the three interrupted isoforms (125,463–125,892); it is reported by the program as having the same donor, branching and acceptor sites for splicing as the KIS68596 transcript (Fig 3A and 3B) and has notably higher expression in the total fraction than in the polyadenylated fraction, with logFC values of approximately 6.5 (Table 1).

Regarding the transcription start sites of the isoforms, it is tempting to say that they share the same site upstream of the start codon of the uncharacterized gene *UMAG_03168*. Read mapping and differential expression analysis shows that the processed isoforms, mainly ter1-i4, could be included among the large quantities of nonpolyadenylated transcripts, which is observed as a nonuniform coverage of reads aligned in the ribodepleted fraction, with a notable increase in a region of approximately 1,280 nt (Fig 3A); and these continuous isoforms are probably reconstructed as a consequence of the existence of both polyadenylated and non-polyadenylated populations of transcripts as in TLC1 [67], that coincide with the patterns observed in stable introns excised from mRNAs [68], so that the increase in abundance suggests that this region corresponds to the stable/mature nonpolyadenylated transcript, and the presence of CE1–CE3 in the reconstructed transcripts (Fig 3A and 3B), reinforces the idea that these elements are surely structural domains. The ter1-i4 could correspond to the precursor transcript of the mature form of TER, this idea and the importance of this particular isoform are reinforced by the notable increase in its expression relative to the levels of various house-keeping genes (Fig 3C).

To confirm the finding of polycistronic-transcript isoforms, a series of primers for reverse transcription and PCR assays were designed, all aligning upstream of the putative Sm site; likewise, to distinguish transcript size changes among polyadenylated mature and non-polyadenylated transcripts, Oligo(dT)$_{18}$ was used to accomplish the reverse transcription from total RNA samples from 521 strain (Fig 4A and S3 Table). We used either Oligo(dT)$_{18}$, random primers, or specific reverse primers WT-ter1-Low and 5ter-rev to synthesize the first cDNA strand, then the subsequent PCR assay were attained using ter-ALL-5a and Template-Low primers produced the expected 469 bp fragment, corroborating the transcriptional activity of the region (Fig 4B and 4B', upper panel). Moreover, despite we did not find an oligo primer-pair quite useful to analyze qRT-PCR isoform abundances, we observe slight decreases in amplification products when Oligo(dT)$_{18}$ was used for reverse transcription; that is why, it was tempting to correlate those results with a poor abundance of polyadenylated TER RNAs as described elsewhere before [67]. Using the ter-i4D-5a forward primer, which was designed to align only the ter1-i4 spliced site, we obtained the expected 637 bp fragments, and no significant amplification was detected when genomic DNA was used as a template. These experiments confirm that the ter1 transcription start site is in the *UMAG_03168 locus* and that the synthesized transcripts are alternatively spliced. When Oligo(dT)$_{18}$ was used for first-strand cDNA synthesis, the amount of amplified DNA was reduced (Fig 4B', lower panel), suggesting that only a minor fraction of the spliced transcripts was polyadenylated; the results back up the transcript would be considered polycistronic in nature.

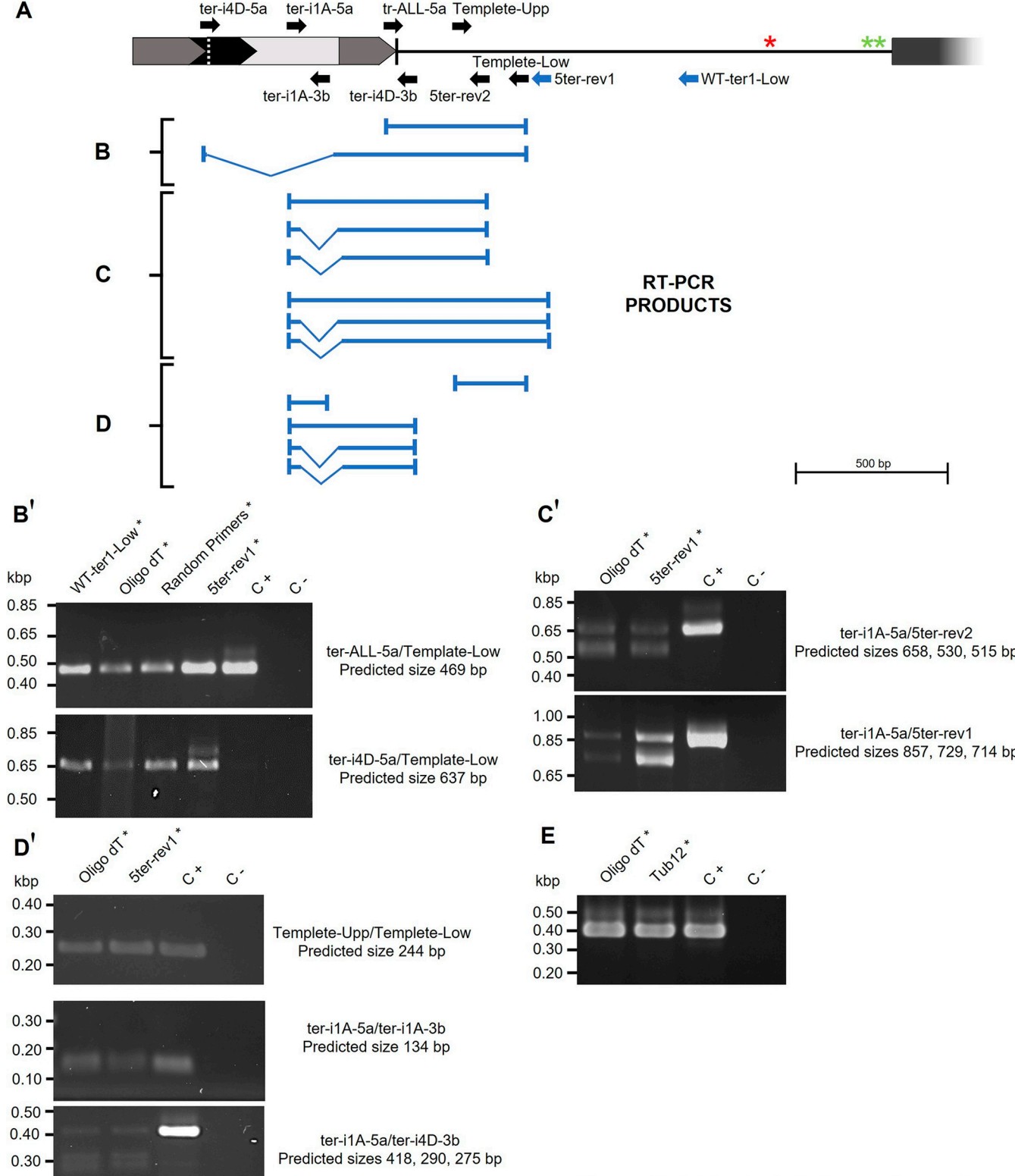

**Fig 4. Confirmation of transcription from the *ter1 locus* occurs.** (A) Upper panel. Schematic representation of *ter1 locus* and alignment sites of primers used for RT-PCR assays located upstream of the putative Sm site (red asterisk) and polyadenylation signals (green asterisks), reverse primers used for cDNA synthesis are represented by blue arrows. (B–D). Combination of primer pairs used and expected amplicon sizes for each combination. (B'–D') Agarose gel

electrophoresis of PCR products obtained from cDNA synthesis with the reverse primers indicated at the top of each lane, genomic DNA was used as a positive control. (B and B'). Confirmation of transcription of the *ter1 locus* containing at least one fragment from the 3' end of *UMAG_03168* and of the existence of the ter1-i4 isoform, evidenced by the presence of a 637 bp fragment unable to amplify in the positive control, top panel, and bottom panel, respectively (C, C' and D, D'). The long ter1-i1 isoform allowed the amplification of the 418 bp fragment with the ter-i1A-5a and ter-i4D-3b primer pair and revealed at least two isoforms processed by alternative splicing. (E) A 405 bp amplified fragment of α-Tubulin transcript was used as a gene expression control.

To bring out additional isoforms produced from the ter1 transcripts, Oligo(dT)$_{18}$ and 5ter-rev1 were used for first cDNA strand synthesis; then, a series of DNA polymerization reactions using swapped primer pairs was achieved (Fig 4C and 4C'). The combination of the forward ter-i1A-5a, which anneals to a ter1-i2 and ter1-i3 left splicing border, and the reverse primers 5ter-rev2 and 5ter-rev1 reveals the existence of at least one spliced isoform within the expected size for ter1-i2 and ter1-i3; again, the polyadenylated fraction seems to be scarce (Fig 4C' bottom panel). The ter-i1A-5a and ter-i1A-3b primer pair is capable of amplifying fragments either from the nonspliced intron of *UMAG_03168* transcripts or from the polyadenylated ter1-i1 isoform; the predicted 134 bp products were amplified when Oligo(dT)$_{18}$ was used for first strand DNA synthesis (Fig 4D' first lane). The same fragment was amplified when specific oligo 5ter-rev1 was used for the first strand synthesis of cDNA, but the predicted fragment could be amplified from both polyadenylated and nonpolyadenylated isoforms ter1-i1 (Fig 4D' second lane). Finally, using the ter-i1A-5a and ter-i4D-3b primer pair, we amplified three cDNA fragments, one with the predicted size for the long ter1-i1 isoform transcribed from the *UMAG_03168 locus* to downstream and two from spliced isoforms with the alternative acceptor sites (Fig 4D' bottom panel).

## *ter1* gene disruption

One-step gene disruption of *ter1* was carried out by homologous recombination in the WT 518 strain using a conventional cassette assembled as described previously. The strategy is depicted in S4A Fig: a 649 bp fragment comprising the template domain was replaced by a 2,045 bp *Bam*HI fragment of pCM1007 containing the chimeric *hph* (S4B Fig). Four hygromycin-resistant transformants were selected, as they recapitulate the features of *trt1Δ* telomerase-negative mutants, i.e., small, dry, slim dark-pigmented colonies with defined, slightly wavy and raised borders [32]. Mitotic stability was reached at ∼72 generations, and total DNA was extracted from each transformant clone, and the replacement of the template sequence was confirmed by PCR assay.

To verify the gene disruption, the oligonucleotide primer-pair WT-ter1-Upp/WT-ter1-Low were employed to amplify the 3,758 bp fragment spanning the selectable marker gene plus adjacent sequences in *ter1*-disrupted strains; a 2,362 bp DNA fragment matching the endogenous WT *locus* was found in strain 518 (S4A and S4C Fig). Stable insertion of the *hph* gene was further confirmed by PCR amplification of 1,757- and 3,004-bp DNA segments spanning from the middle of the *hph* gene to 5' and 3' boundaries, respectively, outside of the flanking sequences used to construct the gene disruption cassette (S4B Fig). The expected sizes were amplified from *ter1Δ* mutants, here after named ter1-02, ter1-24, ter1-35, ter1-40 strains, but not from WT 518 and negative controls (S4C Fig). Finally, using the primer pair Template-Upp/Template-Low, amplification of a DNA fragment of 244 bp comprising the template domain was obtained from the wild-type strain but not from the negative control or disrupted transformants (S4A and S4D Fig). Thus, we conclude that the *ter1 locus* was disrupted.

## Characterization of *ter1Δ* mutants

Once the identity of the selected mutants was verified, the occurrence of progressive telomere shortening was examined by extracting DNA samples from a series of continuous subcultures. Every sample was taken after approximately 24 duplication rounds from the prior and used to analyze the terminal restriction fragment (TRF) of chromosomes. As seen in Fig 5A, we

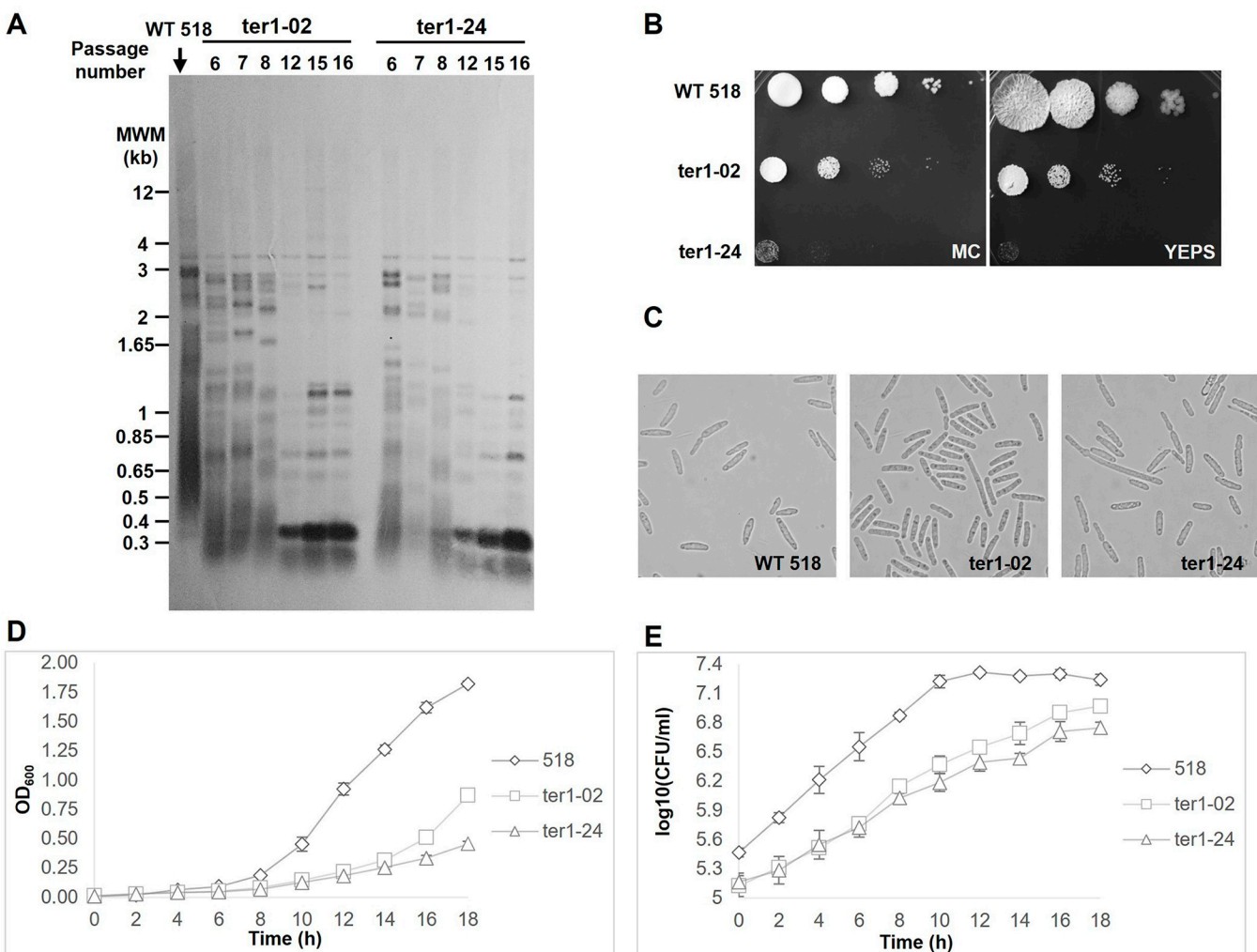

**Fig 5. Phenotypic alterations caused by the interruption of *ter1* in *U. maydis*.** (A) TRF analysis of the WT 518 strain and ter1 mutants. The loss of TER triggers a progressive shortening of TRF and eventually leads to the emergence of surviving cells showing the amplification of a 350 bp fragment. (B) Serial 10-fold dilutions of cultures were spotted on YEPS or MC plates and incubated for 2 days at 28°C; ter1 mutants showed dramatic decreases in colony formation and colony size, consistent with the loss of telomerase activity described in other organisms. (C) Cell morphology of strain WT 518 and mutants ter1-02 and ter1-24. The interruption of *ter1* causes alterations in cell morphology, and the mutants have irregularly edged cells and an increase in the presence of elongated cells with alterations in the budding angle, suggesting defects in cell cycle regulation. (D and E) Growth kinetics of the WT strain and ter1 mutants. Strains were grown in YEPS broth (28°C) and sampled every two hours. Samples were used for measurements of $OD_{600}$ or were diluted and plated. Colonies were counted after 24 h of incubation at 28°C and plotted. The results of triplicate assays are shown.

compared wild-type vs. mutant strains at approximately 108 replication rounds (lanes 1 and 2). Both samples corresponded to a 6th subculture. At that timepoint, in the mutant strain, telomeres were noticeably shortened compared with those of the control. Samples were also taken at the 7th and 8th subcultures (132 and 156 replication rounds, respectively; Fig 5A, lanes 3 and 4, respectively), and at replication round 252, a *Pst*I DNA fragment of 350 bp that matched the *rumT* repeat plus short telomere sequences could be amplified from Type I survivors; in addition, radiography revealed stronger hybridization signals from sequences of ~700 bp and ~1,200 bp that could correspond to multiple series of *rumT* or even rearrangements of *UTASa* sequences (currently under study).

The growth kinetics of the two survivor strains were calculated as described previously. The growth of sporidia in culture in at least three batches was plotted as $OD_{600}$ vs. time and as

CFU/ml vs. time (Fig 5D and 5E). A significant (p = 0.05) delay in cell proliferation was noted in the ter1-2 and ter1-24 cultures in comparison with 518 cultures; the wild-type strain reached stationary phase after 12 h at 30˚C, whereas the ter1-02 and ter1-24 mutants continued to replicate (Fig 5D). When $\log_{10}$ (CFU/ml) was plotted vs. time, the decrease in cell viability (p = 0.01) from the start point was clearer in both disrupted mutants than in strain 518 (Fig 5E). Taking into account the $\log_{10}$ (CFU/ml) vs. time from 2 h to 12 h, linear regression analysis gave $R^2$ > 0.98, allowing us to determine the generation times in our experiments as 117 min for strain 518, 176 min for strain ter1-02, and 205 min for strain ter1-24. The combination of growth delay and deficiency in replicative potential should cause a dramatic diminishing of the colony appearance. Indeed, after 1:10 serial dilutions of cultures grown to OD = 1.0, differences in both the number and size of the colonies were observed for both mutants, with the most dramatic diminution of growth potential observed in the ter1-24 strain, regardless of the medium used (for example, YEPS or MC) (Fig 5B). The cellular morphology of strain 518 and in its two *ter*-disrupted mutants, ter1-02 and ter1-24, was also examined under a light microscope. Mutants showed elongated sporidia with slender irregular contour, vacuolation and, to a lesser extent, alterations in the growth polarity and angle of the bud and mother cells (Fig 5C). The elongated shape and perpendicular buds suggested to us that cell cycle activity was defective in these telomerase-negative *ter1Δ* mutants. Other infrequent alterations could also be observed.

## Loss of the *ter1* gene also causes chromosome instability

The electrophoretic karyotype of wild-type strain 521 was obtained and compared with that of both types of telomerase-negative mutants of *U. maydis*. Whereas the TERT mutant trt1-1 (Type I survivor) showed increased chromosome size possibly to extensive TAS + telomere overamplification (currently under study), the trt1-2 (Type II survivor) exhibited moderate lengthening of some chromosomes and visible rearrangements of others (Fig 6A). On their own, TER mutants gave rise to only Type I survivors; of these, ter1-02 shows gross chromosomal rearrangements or even extensive smearing such that individual chromosome bands were poorly distinguishable; the ter1-24 karyotype was very similar to wild-type strain (Fig 6A). Additionally, the cells were stained with propidium iodide and calcofluor white as described in the Materials and Methods and examined under optical and epifluorescence microscopy (Labophot-2, Nikon, Japan). The staining of nuclei with propidium iodide also allowed us to record the percentages of budding sporidia at 90, 45 and 0˚ as well as the percentage of nuclear aberrations occurring in telomerase-negative *trt1*-disrupted and *ter1Δ* strains compared with the same events occurring in wild-type cells. The results are plotted in Fig 6B and 6C. As in *trt1Δ* mutants, we observed two phenotypes that, as will be discussed, are related to the cells' telotypes. The WT sporidia showed a rod-like shape (13 μm x 3 μm) with a smooth contour, budding at ∼30–45˚ from the mother length axis, and poor vesicle content, as has been described elsewhere [69]. The *ter1Δ* cells included a significantly elevated percentage (p = 0.5) of lengthened and mononucleated cells (Fig 6B) and even septate sporidia, with some septum lacking nuclei, which suggested possible cell cycle arrest or even nondisjunction of chromosomes. The TERT-disrupted mutants described by Bautista-España *et al*. [32] showed elongated cells, with minor disturbances in budding angle compared with *ter1Δ* mutants (Fig 6B), while ter1 mutants show an increase in the formation of nuclear aberrations and chromosome bridges (Fig 6C).

## Plant infection assays

The time course of the infection in maize seedlings was monitored for 21 days, and the results are shown in Fig 7A. In both crosses, chlorosis appeared in the leaves, and at approximately 5

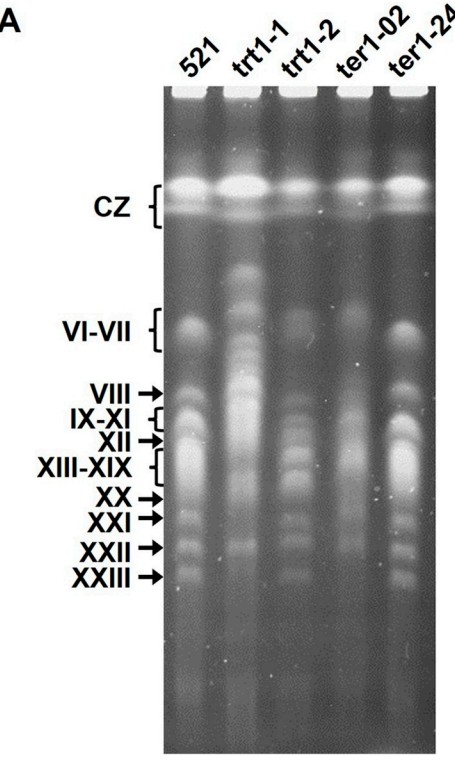

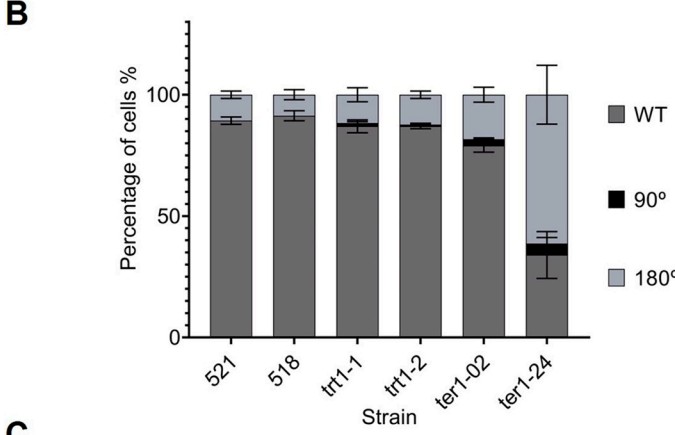

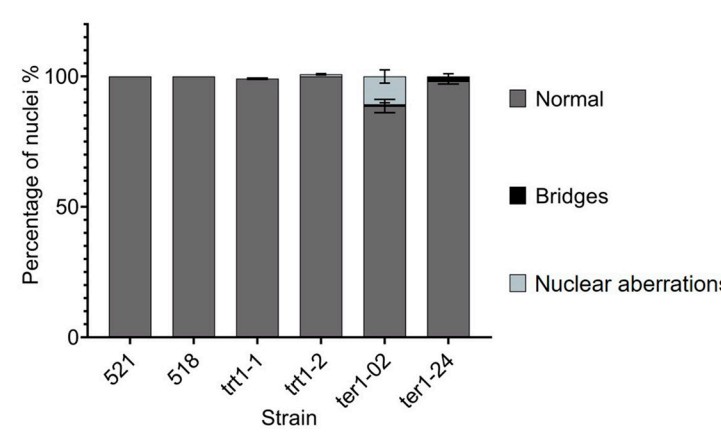

**Fig 6. Electrophoretic karyotype and percentage of cells exhibiting nuclear aberrations and alterations in the budding angle.** (A) Electrophoretic karyotype of WT 521 strain and four telomerase negative mutants trt1-1 and trt1-2 are TERT-disrupted mutants (*trt1 Δ*) and ter1-02 and ter1-24 are TER-disrupted mutants (*ter1Δ*); trt1-1 and ter1-02 strains (lanes 2nd and 4th respectively) show increases in chromosome size, possibly due to *UTASa* amplification, both strains also exhibit fuzzy bands along the entire lane possibly associated to DNA DSBs, which coincided with the formation of nuclear aberrations; in contrast, the trt1-2 and ter1-24 strains had a similar karyotype to that of the WT strain. (B) *trt1*-disrupted mutants also show slight changes in the budding angle, which in the *ter1*-disrupted strains is noticeable, mainly as polar growth, perhaps as consequence of alterations in cell cycle regulation. (C) Interruption of *ter1* triggers an increase in the formation of nuclear aberrations and chromosome bridges, suggesting DNA damage response system; in contrast, *trt1* mutants exhibit an only slight increase in the formation of these structures.

dpi, chlorosis percentages of 48%, 36% and 37% were recorded for the positive control and for heterozygous crosses with strains ter1-02 and ter1-24, respectively (Fig 7A). At 21 dpi, 89% of the controls presented symptoms of infection, in contrast to 45% and 47% for heterozygous *ter1⁻* crosses, in which only the appearance of chlorosis was observed, accompanied in some cases by a slight twisting of the leaves (Fig 7A and 7B). The anthocyanin formation during the development of the infection was infrequent and subtly perceptible in the heterozygous crosses (Fig 7D). Tumor development was not observed, and approximately 47% of plants in both cases did not develop symptoms of infection, in contrast to the plants inoculated with the crosses of wild strains, which at that time showed tumors developed on all the aerial tissues of the plants, 35% of which corresponded to mature tumors with teliospore content (Fig 7A and 7E). Thus, it is concluded that TER plays a crucial role in the development of *U. maydis* infection and the completion of the life cycle.

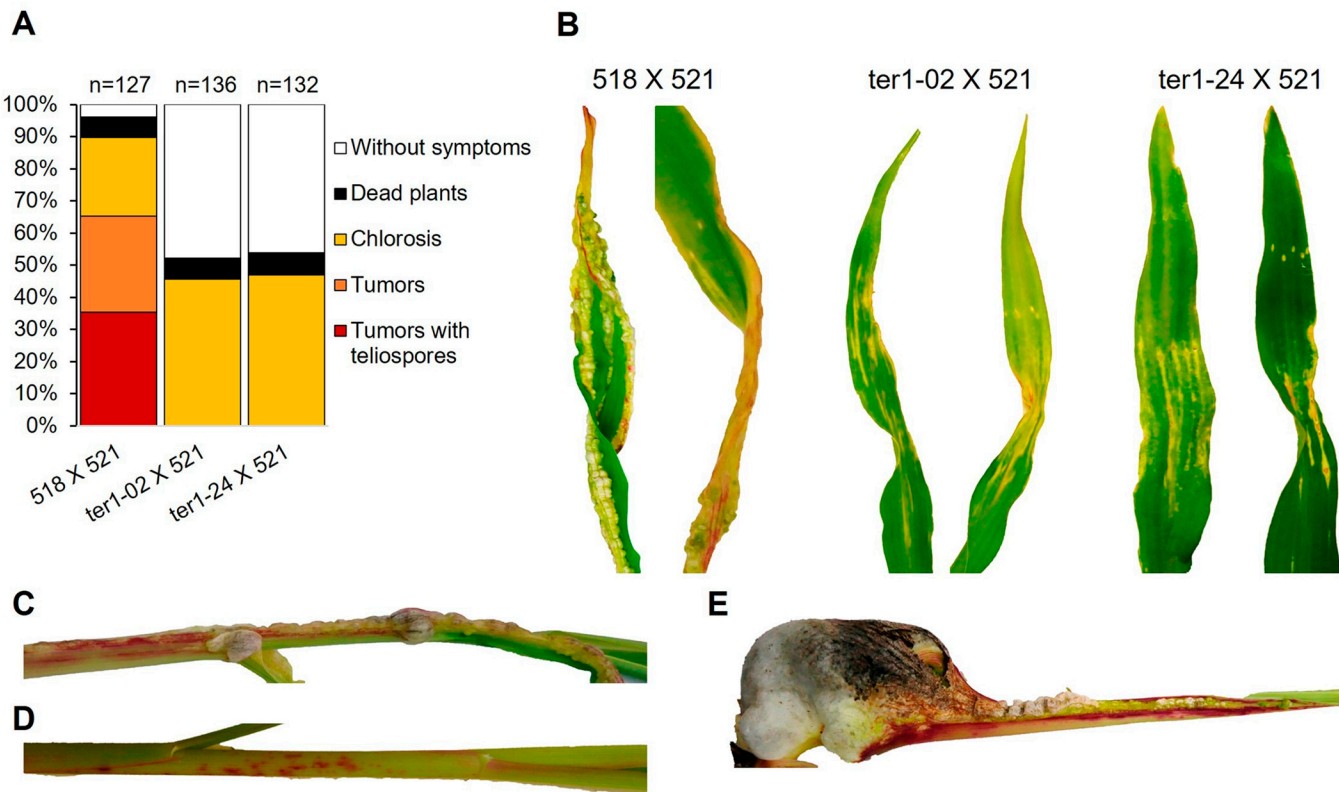

**Fig 7. Development of infection in maize plants.** (A) Quantification of symptoms in maize plants infected with the indicated crosses at 21 dpi. The mean values of three independent experiments are shown. (B) Details of the leaves and the stem at 21 dpi of the plants inoculated with the wild crosses and heterozygous crosses *ter1⁺/ter1⁻*. (C and D) Stem detail and anthocyanin formation at 21 dpi of plants inoculated with the wild crosses and heterozygous *ter1⁺/ter1⁻* crosses, respectively. (E) tumor developed by wild crosses at 21 dpi.

## Discussion

In this work, we identified the region containing the TER gene in reported species of the Ustilaginales though the mining of a series of template sequences complementary to the TTAGGG telomeric repeat of *U. maydis* [30] and comparative genomics. We found a single copy of the putative telomerase RNA gene in all the examined basidiomycetes. This approach was modeled after the bioinformatic approach previously described by Chakrabarti *et al.* [25]; however, this strategy can be limited by the availability of sufficient genome sequences from the relatives of the species of interest. Although the molecular approach has given excellent results in the identification of the TER subunit of several organisms, that strategy requires more complex methods and expensive consumables than the present method [70,71]. To identify the TER subunit, it was essential to recognize the conserved structural domains important for telomerase function [12,24]; here, the main structural domain we sought in the syntenic region was the template/pseudoknot domain, which was preserved throughout the taxonomic group and fulfilled the basic requirements to be considered analogous to those reported elsewhere [72].

The template domain of *U. maydis* TER is 9 nt long, unlike that of other members of the Ustilaginales, which have a length of 11 nt (Fig 2B), but this difference does not cause any apparent disturbance of telomerase processivity, as telomeric repeats of *U. maydis* are evenly synthesized in all the samples to date reported [30,33]. Downstream of the template domain, uracil-rich sequence stretches continued by adenine-rich sequence stretches resembling sequences CS3 and CS4 were found conserved in all the examined Ustilaginales species suggesting a pseudoknot domain located between the template and Sm-like domains, analogous to that found of yeasts and filamentous fungi [73–75]. In *U. maydis*, this region is reasonably conserved with some indels between residues; in such important sequence, the presence of A/U bases could facilitate the formation of U-A•U interfaces that could contribute to catalytic activity [73,76]. Accordingly, we propose that bases 126,810 to 126,840 (irp 730–760) make up stem 1 and loop 1 of the pseudoknot domain, while the poorly preserved region 126,841 to 126,851 (irp 761–771) could form loop 2, resembling the poorly conserved sequence stretches which were found to be dispensable in hTR and TLC1 [77–79], while, the stretch from 126,852 to 126,860 (irp 772–780) is part of stem 2 (S2C Fig).

After findings of Chen *et al.* [59] and Kim *et al.* [60] on mTR and hTR, respectively, it is attractive to propose a P6.1-like helix structure toward the 3'; it could be feasible by their location, and conserved sequence along with the existence of U residues which could serve as target of possible modifications (S1D Fig). Downstream, the AAU$_{5-6}$CY sequence corresponds to a hypothetical Sm7-binding site for the Sm7 complex, which in *S. cerevisiae* is required for the stability and processing of TLC1 [70,80]; similar sequences are found in fungal TERs [73–75]. This putative domain spans the minimum consensus sequence in the fungi here studied, but in some ascomycete species, such as *Aspergillus* spp. this motif is extended by a pair of uridine residues toward the 3' end and contains intercalated cytosine residues. As in the members of the Ustilaginales, terminal purine residues are absent (S1E Fig), suggesting flexibility in the Sm site, as was previously reported from its tolerance to induced mutations [81].

We did not find canonical splicing sites along the intergenic region where *ter1* lies; however, potential polyadenylation sites downstream of the hypothetical Sm7-binding were found, suggesting the endpoint of TER was nearby. It remains to be experimentally determined whether other processes, such as alternative splicing, occur at either the 5' or 3' end of the transcript for its maturation. Our findings, which arose from transcriptome analysis from both ribo-depleted nuclear RNA and total mRNA, were supported by the finding of key DNA stretches amplified by RT-PCR from the predicted transcripts and their isoforms (Fig 4) which

matched with ESTs deposited at dbEST database (accession numbers: JK215994, JK215995, and JK842063) and the transcriptome analysis reported by Donaldson *et al*. [66].

Existence of isoforms assembled from RNAseq data, was confirmed using Oligo(dT)$_{18}$ or specific oligonucleotide primers to synthesize the first strand of cDNA. RT-PCR using Oligo (dT)$_{18}$ render, most of the times faint fluorescent signals form cDNA bands stained with ethidium bromide, whereas cDNA synthesized with specific reverse primers were intense. Although no quantitative assay was performed, the results suggested that the mature version of ter1 could be derived from the polyadenylated polycistronic ter1-i1 isoform, seen as the fraction of collapsed reads in the bottom panel of Fig 3. That fraction had a similar length ($\approx 1,280$ nt) to the transcript observed by Chen *et al*. (1,291 nt, personal communication, accession number ON015423), in addition to the end matching with the position of the putative Sm$_7$-binding site. Our results suggest that unlike TLC1, where the polyadenylated and nonpolyadenylated fractions are not processed by separate pathways [67,82], in *U. maydis* the former is the precursor of the mature version. Moreover, the presence of orthologues of the Sm7 complex proteins, within the fungal genome (*UMAG_10381*, *UMAG_04781*, *UMAG_11043*, *UMAG_12244*, *UMAG_10312*, *UMAG_12130* and *UMAG_10805*) and hypothetic genes homologous to Mex67 and Mtr10 involved in TLC1 biogenesis in yeast were found in *U. maydis* (*UMAG_01688* and *UMAG_01670* respectively), hinted a functional role of the Sm7 complex in the maturation of the ter1 subunit of telomerase, as occurs in yeast [80,83]. Lack of information for termination mechanisms of snRNAs transcription in *U. maydis* comparable to those of yeasts [84] also hinder the role of the two polyadenylation signals predicted downstream the putative Sm$_7$-binding site of *ter1*. Transcriptome sequencing of cap-captured RNA, or one-molecule RNA sequencing, could give us additional insights.

Disruption of the *ter1 locus* led to a phenotype of slow growth, poor proliferation potential, telomere attrition, changes in colony morphology (to small and slightly pigmented colonies) analogous to those occurring in telomerase-negative mutants [32,85]. However, *ter1Δ* mutants exhibited cellular abnormalities such as uninucleate cell elongation, changes in the budding angle and alterations in polar growth at higher percentages than their *trt1* counterparts (Fig 6B and 6C). Those abnormalities prompted us to perform PFGE analysis; smearing of the DNA along the size range of the *U. maydis* chromosomes suggested the occurrence of chromosomal fusion and ruptures, and that cell elongation was a result of telomeric dysfunction and impaired chromosome segregation, as has been reported previously [86–88]. However, it is necessary to establish whether the formation of type I and type II survivors is differentially affected when *ter1* or *trt1* is disrupted; those data could give us insight about the roles of the subunits in the expression of additional factors and mechanisms differentially controlled by each of the subunits. Abrogation of DDR could have impaired cell cycle progression [89–91] and caused the slight differences observed between the two mutants.

The failure of mutants to develop tumor galls when crossed with the wild-type 518 strain *in planta* is currently under research; however, recent reports have shed light on the role of telomere length in fertility and embryogenesis in humans and mice [92,93]. Further studies could give more insight on the requirements of telomere length or telomerase components on teliospore formation. Extensive transcriptome analyses, imaging, karyotyping, and more are needed to fully characterize *U. maydis* telomerase-negative mutants before the usefulness of this model system for telomere analysis is fully realized.

## Conclusions

As previously stated, we found the TER subunit of the *U. maydis* telomerase by a computational method and characterized it both as a functional gene and as the display of transcript

isoforms derived from it. The *ter1* gene was disrupted in the strain 518, the complementing strain of the 521, with expectations on future studies. However, the functionality of the TER domains identified here requires further analysis. The preservation of these domains within a gene that is divergent in length and sequence provides an adequate model for studies of the function and evolution of the domains that make up the TER subunit. The observation of the phenotypes in *ter1Δ* also highlights the importance of employing alternative models in which abrogation of these components is feasible and can provide new information on their contribution to cellular development. Lastly, the life cycle and genotypic differences between Ustilaginales species are useful for exploring the possible processing pathways and extratelomeric functions of this component within dimorphic organisms.

## Supporting information

**S1 Fig.** A. Multiple alignment of intergenic sequences where the CE1, CE2, and CE3 domains of the gene encoding the TER subunit are located in boxes. B. Multiple alignment of intergenic sequences of the gene encoding the TER subunit in the section boxing the putative template domain. C. The multiple alignment of intergenic sequences shows the pseudoknot-like domain located in a box on the putative genes encoding the TER subunit. D. Multiple alignment of intergenic sequences showing the P6.1-like domain of the gene encoding the TER subunit in a box. E. Sm-binding site-like domain boxed on the multiple alignment of intergenic sequences of the gene encoding the TER subunit. F. Final segments of intergenic sequences of the multiple alignment of the putative genes encoding the TER subunit in several Ustilaginales.
(TIF)

**S2 Fig. Folding of conserved structural domains in Ustilaginales species.** (A) MSA of the conserved region reminiscent of the P6.1 sequence in vertebrates. (B) Secondary consensus structure of the P6.1 hairpin according to the RNAalifold program. Nucleotides conserved in all species are in red, and nucleotide transitions among species are in blue; base pairings supported by covariations are marked by asterisks. (C) Structure of the pseudoknot domain. Potential folding of the conserved region resembling CS3 and CS4. Red nucleotides represent the conserved residues in at least 80% of the species.
(TIF)

**S3 Fig. Splicing sites identified within *UMAG_03168* and the polyadenylation site of *ter1*.** (A) WebLogo of the splicing sites of *UMAG_03168*. D1 corresponds to the sequence of the donor points used in the processing of KIS68596 and ter1-i4, D2 corresponds to the donor site used for the processing of isoforms 2 and 3 of *ter1*, and B corresponds to the branch site sequence used in the processing of the different isoforms observed. (B) Multiple alignment of the branch point location and acceptor sites used in *U. maydis* for the processing of transcripts. The pairs of AG bases used as acceptor sites are highlighted in purple, showing that the alternative acceptor site used in the processing of ter1-i3 is located near a region with AG residues conserved across Ustilaginales species. (C) Multiple alignment of the 3' end of the *ter1 locus*. Final region of the *ter1 locus* where a possible polyadenylation signal was identified, which is located in a region that is rich in A residues in Ustilaginales species.
(TIF)

**S4 Fig. Strategy for confirmation of the integration by HR and PCR assays to verify the interruption of the *locus*.** Target sequence deletion was tested by PCR assays in which different pairs of primers were designed for the WT strain and interrupted transformers. (A) *locus* interruption strategy and alignment diagram of the primers designed to check the integrity of

the *locus* in the WT strain. (B) Diagram of the alignment of the primers to verify the replacement of the target sequence with the *hph* gene sequence. (C) The replacement of the *locus* was analyzed in 4 transformants: ter1-02, ter1-24, and ter1-35 and ter1-40. They show an increase in the size of the amplified size corresponding to the size of the replacement of the target sequence by the gene *hph*. The WT strain produces the expected amplicon of 2,362 bp, whereas the negative control does not produce any amplification. The integration of the selection marker sequence was checked at both ends of the recombination site using pairs of primers aligning within the *hph* gene sequence and outside the recombination site, and amplification products of the expected size were obtained in the *ter1* mutants. In the integration check from the 5' end, the expected amplification product of 1,757 bp was obtained from mutants ter1-02, ter1-24, ter1-35, and ter1-40. In the integration check from the 3' end, the expected amplified product of 3,004 bp was obtained from mutants ter1-02, ter1-24, ter1-35, and ter1-40. Amplification was not obtained from the WT strain or negative controls. (D) Corroboration of the absence of the target sequence. Pairs of primers aligning within the deleted region were used. A fragment of 244 bp corresponding to the expected size was obtained in the WT strain, but no amplification products were obtained from the interrupted mutants ter1-02, ter1-24, ter1-35, and ter1-40 or from the negative control.
(TIF)

**S1 Table. Location of *ter1 locus* within Ustilaginales species.**
(XLSX)

**S2 Table. Analysis of DEGs around the *ter1 locus* obtained with the EdgeR method.**
(XLSX)

**S3 Table. Primers used in this study.**
(DOCX)

**S1 Raw images. Agarose gel electrophoresis of PCR amplified products and blot assays.**
(PDF)

## Acknowledgments

To CINVESTAV-IPN Irapuato staff, especially Plinio Guzmán Villate for his teach and for initiating this research line, and to José Ruíz-Herrera and Alfredo Herrera-Estrella for their kind comments on our research; to W.K. Holloman, always by his teach, advice and comments have made on our work.

## Author Contributions

**Conceptualization:** Patricia Sánchez-Alonso.

**Data curation:** Juan Antonio Sanpedro-Luna, Leticia Vega-Alvarado.

**Formal analysis:** José Juan Jacinto-Vázquez, Leticia Vega-Alvarado, Patricia Sánchez-Alonso.

**Investigation:** Juan Antonio Sanpedro-Luna, José Juan Jacinto-Vázquez, Estela Anastacio-Marcelino, Carmen María Posadas-Gutiérrez, Candelario Vázquez-Cruz.

**Methodology:** Estela Anastacio-Marcelino, Carmen María Posadas-Gutiérrez, Moisés Carcaño-Montiel, Leticia Vega-Alvarado, Candelario Vázquez-Cruz.

**Resources:** Moisés Carcaño-Montiel, Candelario Vázquez-Cruz.

**Software:** Iván Olmos-Pineda, Jesús Antonio González-Bernal.

**Supervision:** Candelario Vázquez-Cruz, Patricia Sánchez-Alonso.

**Validation:** Estela Anastacio-Marcelino.

**Visualization:** Juan Antonio Sanpedro-Luna, Estela Anastacio-Marcelino.

**Writing – original draft:** Juan Antonio Sanpedro-Luna.

**Writing – review & editing:** Patricia Sánchez-Alonso.

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
