## [Decision Letter · Decision Letter 0]

21 Nov 2022

PONE-D-22-27372Telomerase RNA plays a major role in the completion of the life cycle in Ustilago maydis and shares conserved domains with other UstilaginalesPLOS ONE

Dear Dr. Sanchez-Alonso,

Thank you for resubmitting your manuscript to PLOS ONE. After careful consideration, we feel that it has merit but does not fully meet PLOS ONE’s publication criteria as it currently stands. Therefore, we invite you to submit a revised version of the manuscript that addresses the points raised during the review process.

An additional issue has been raised by  Reviewer 2 that was not addressed in a previous critiques. Specifically is not fully clear that that the gene identified does indeed encode the telomerase RNA. This should be accomplished, if technically feasible, by a simple complementation assay. In the absence of this information, a qualification of the conclusion must be made within the paper and in the title.

In addition to this experimental change, please address minor issues raised by the second reviewer.

We look forward to receiving your revised manuscript.

Kind regards,

Arthur J. Lustig, PhD

Academic Editor

PLOS ONE

Journal Requirements:

Reviewers' comments:

Reviewer's Responses to Questions

**Comments to the Author**

1. Is the manuscript technically sound, and do the data support the conclusions?

Reviewer #1: Yes

Reviewer #2: Partly

2. Has the statistical analysis been performed appropriately and rigorously? 

Reviewer #1: Yes

Reviewer #2: Yes

3. Have the authors made all data underlying the findings in their manuscript fully available?

Reviewer #1: Yes

Reviewer #2: Yes

4. Is the manuscript presented in an intelligible fashion and written in standard English?

Reviewer #1: Yes

Reviewer #2: Yes

5. Review Comments to the Author

Reviewer #1: This revised manuscript reports the identification and characterization of the gene encoding telomerase RNA from Ustilago maydis, a basidiomycete fungus. The RNA is evidently synthesized from an upstream ORF (UMAG_03168), and subjected to splicing and processing to generate the mature form. Disruption of the telomerase RNA gene results in telomere shortening and reduced proliferation. The ter1 mutant is also defective in several aspects of the U. maydis life cycle. The authors identified conserved features of the RNA, including potential pseudoknots and Sm-binding sites.

The discovery of telomerase RNA from a basidiomycete fungus is an important advance and will pave the way for structural, functional and comparative analysis of this conserved lncRNA. The main concern for the original manuscript was the lack of strong supporting data for the proposed mechanism of RNA biogenesis involving transcription from an upstream protein coding RNA. The authors have now provided RT-PCR analysis that are consistent with the proposed mechanism. They also made additional changes to address other concerns. I believe these changes have substantially strengthened the manuscript.

It is worth noting that during the revision, the Julian Chen lab published a PNAS paper that describes their characterization of the same telomerase RNA. I think the two studies are complementary. The Chen paper provides a very thorough and convincing characterization of the biogenesis mechanism that leaves very little doubt about the nature of the precursor as a polycistronic RNA. The current manuscript, on the other hand, provides interesting data on the role of this RNA in vivo; the telomere lengths analysis of the ter1 mutant is more convincing and the role of this RNA in the life cycle of U. maydis is a nice finding. In short, I think this is a solid manuscript that helps to advance the field.

Reviewer #2: Regarding the manuscript titled "Telomerase RNA plays a major role in the completion of the life cycle in Ustilago

maydis and shares conserved domains with other Ustilaginales" by Sanchez-Alonso et al. I think the authors have appropriately addressed the concerns of the previous reviewers however I have one concern and the editor will need to determine if it must be carried out before publication This is, that authors did not confirm that the deletion they carried out was responsible for the observed phenotypes. Generally a deletion mutant is complemented by adding back the gene at an ectopic locations. In this case the authors would need to add back the RNA and determine if doing so complements the deletion phenotypes observed. This could be done using genome integration or an autonomously replicating plasmid and assessing the haploid cell phenotypes. If this is not done i think the title should be modified to indicate that the gene identified is a probable or predicted telomerase RNA.

Other minor points the authors may consider addressing are wording issues and matters of sentence clarity. I have highlighted some below

Lines 117-121 Clarify, when does the repression of the putative TER gene occur perhaps break the sentence into two to help clarify the points being presented

Line 126 the phrase "order representatives here studied" is awkward and not clear, edit for clarification

Lines 237-239 After running the sucrose cuchion centrifugation were samples assessed to determine the purity and degree of subcellular fractionation, ie how pure were the nuclei isolated in this manner?

Lines 550-554 The portion of the sentence beginning "ratifying ..." is not clear and could be edited to improve clarity. Also consider breaking up this overly long sentence

Lines 581-582 "results points to the fact" should be edited to results suggest or results support an RNA maturtion process that involves

Lines 583-586 The portion of the sentence " ...,so unlike TLC1... " seems like discussion not results. Consider moving this to the discussion.

Line 612 the phrase "every at approximately 24 duplication rounds, " is not clear edit to clarify what you intend to convey

Line 649 and elsewhere The phenotypic differences that distinct mutants present lead me to ask - Were there any DNA-level differences detected between the mutants that might help explain mutant to mutant variation?

Line 687 has should be had

Line 751 should " extended a pair" be extended by a pair?

Lines 752-754 wording of this sentence could be improved eg denotate does not seem to be a good word choice here

Lines 757-759 the sentence beginning "Whether ..." is not clear and could be edited to improve clarity

Lines 796 -798 This sentence is not clear edit to better explain what you are trying to convey

6. PLOS authors have the option to publish the peer review history of their article (what does this mean?). If published, this will include your full peer review and any attached files.

Reviewer #1: No

Reviewer #2: **Yes: **Barry Saville

---

## [Author Response · Author response to Decision Letter 0]

28 Dec 2022

To the Editor:

Thank you and to the reviewer for the time and dedication spent in the review of this paper; we acknowledge all the work.

We have changed all the points important for the publication of the work; we included the raw images in Supporting Information at the moment. We hope is a good option.

To Prof. Dr. Barry Saville.

In relation to ter1Δ complementation, it is technically unfeasible to do so because of the great lability of this kind of survivor; in our hands, the frailty of ter1Δ survivors is deeper than that of their trt1Δ counterparts. The additional challenges to do so relied in the deregulation of genes involved in the cell wall composition (among others) that practically impede the obtention of living protoplasts to achieve gene transformation; mutant cells also die after prolonged times of incubation in cell wall lysing enzymes (which are toxic). Protoplast/DNA fusion with PEG also requires temperature changes that ter1Δ mutants are feeble to endure, and finally, the replication times are large. We could assume 8 hours of incubation in rich medium to start the chimeric ter1 expression in transformants; this is true in wild-type cells, but long days would be required to obtain one or no transformants after several experiments.

However, we follow the work of Logeswaran et al., 2022. Proc. Natl. Acad. Sci. USA to reinforce our finding of the telomerase RNA subunit of Ustilago maydis.

Regarding the other minor points the authors may consider addressing, which are wording issues and matters of sentence clarity, and you gently have highlighted, they have been reviewed.

Lines 117-121 Clarify, when does the repression of the putative TER gene occur perhaps break the sentence into two to help clarify the points being presented

It has been edited; thank you for the suggestion.

Line 126 the phrase "order representatives here studied" is awkward and not clear, edit for clarification

It has been edited; thank you for the advice.

Lines 237-239 After running the sucrose cuchion centrifugation were samples assessed to determine the purity and degree of subcellular fractionation, i.e., how pure were the nuclei isolated in this manner?

Nuclei were harvested in a small fraction as described in the text and treated as described by Groner and Phillips (1975). The nuclei were recognized in a small sample stained with propidium iodide under a fluorescence microscope. Staining was performed as described in Alfa et al., 1993. In: Experiments with fission yeast. A laboratory course manual. Experiments 1 and 2. CSHL Press NY USA. Nuclei were harvested with mitochondria; however, after total RNA extraction and ribodepletion of the nuclei fraction, the sequenced transcriptome was useful for this work, as it had the read number required to publish after filtering for mitochondrial transcripts. Because no quantitative comparisons were made against total cytoplasmic or total cellular RNA, no highly purified nuclei were required in this work.

Lines 550-554 The portion of the sentence beginning "ratifying..." is not clear and could be edited to improve clarity. Additionally, consider breaking up this overly long sentence

It has been edited; thank you for the suggestion.

Lines 581-582 "results points to the fact" should be edited to results suggest or results support an RNA maturtion process that involves.

It has been edited; thank you for the suggestion.

Lines 583-586 The portion of the sentence " ...so unlike TLC1... " seems like discussion not results. Consider moving this to the discussion.

The text fragment has been moved to lines 766-770; thank you for the suggestion.

Line 612 the phrase "every at approximately 24 duplication rounds, " is not clear edit to clarify what you intend to convey

It has been edited to clarify; thank you for the suggestion.

Line 649 and elsewhere the phenotypic differences that distinct mutants present lead me to ask - Were there any DNA-level differences detected between the mutants that might help explain mutant to mutant variation?

There is a point mutation in the same locus that is under study; perhaps is speculation, however, we require complete genome sequencing of the strains to discard any other external effects. As stated by Chen et al. and us, those loci are rich in (apparently functional) transcript isoforms.

Line 687 has should be had

It has been edited; thank you for the advice.

Line 751 should " extended a pair" be extended by a pair?

It has been corrected; thank you for the advice.

Lines 752-754 wording of this sentence could be improved eg denotate does not seem to be a good word choice here.

The text has been edited; thank you so much.

Lines 757-759 the sentence beginning "Whether ..." is not clear and could be edited to improve clarity

Your suggestion has been addressed. Thank you

Lines 796-798 This sentence is not clear edit to better explain what you are trying to convey

It has been edited; thank you for the suggestion.

We appreciate your time and dedication to this work.

Thank you sincerely,

Dr. Patricia Sánchez Alonso

Corresponding Author

---

## [Decision Letter · Decision Letter 1]

19 Jan 2023

Telomerase RNA plays a major role in the completion of the life cycle in Ustilago maydis and shares conserved domains with other Ustilaginales

PONE-D-22-27372R1

Dear Dr. Sanchez-Alonso,

We’re pleased to inform you that your manuscript has been judged scientifically suitable for publication and will be formally accepted for publication once it meets all outstanding technical requirements.

Kind regards,

Arthur J. Lustig, PhD

Academic Editor

PLOS ONE

Additional Editor Comments (optional):

Reviewers' comments:

Reviewer's Responses to Questions

**Comments to the Author**

1. If the authors have adequately addressed your comments raised in a previous round of review and you feel that this manuscript is now acceptable for publication, you may indicate that here to bypass the “Comments to the Author” section, enter your conflict of interest statement in the “Confidential to Editor” section, and submit your "Accept" recommendation.

Reviewer #2: (No Response)

2. Is the manuscript technically sound, and do the data support the conclusions?

Reviewer #2: Yes

3. Has the statistical analysis been performed appropriately and rigorously? 

Reviewer #2: Yes

4. Have the authors made all data underlying the findings in their manuscript fully available?

Reviewer #2: Yes

5. Is the manuscript presented in an intelligible fashion and written in standard English?

Reviewer #2: Yes

6. Review Comments to the Author

Reviewer #2: I would like to thank the authors for the respectful response to my initial concerns regarding the manuscript. While I accept that the ter1Δ complementation would be unfeasible in the traditional sense, that is growing up the mutant and adding back the wildtype gene, the authors should consider a different approach. Perhaps in a follow on experiment, this approach is a two step deletion in which they insert a wildtype copy of the locus to be deleted into the IP locus under the control of a promoter that can be controlled eg by carbon source then they delete the wildtype copy of the gene and grow out the deletion mutants with hyg selection and under permissive conditions expressing the ectopically inserted gene. Subsequently they can double spot the double transformants under permissive and non-permissive conditions and assess the impact on phenotypes previously assessed. We have gotten this to work with genes that are lethal when deleted. The advantage of considering this is it would allow mutational analysis of the ectopically expressed copy. I think this might take some time and I think it may open up the possibility for future investigations so I would not consider this a requirement for publication of this work, but it may be interesting for future analyses.

7. PLOS authors have the option to publish the peer review history of their article (what does this mean?). If published, this will include your full peer review and any attached files.

Reviewer #2: **Yes: **Barry J Saville

---

## [Editor Report · Acceptance letter]

15 Mar 2023

PONE-D-22-27372R1 

Telomerase RNA plays a major role in the completion of the life cycle in *Ustilago maydis* and shares conserved domains with other Ustilaginales 

Dear Dr. Sánchez-Alonso:

I'm pleased to inform you that your manuscript has been deemed suitable for publication in PLOS ONE. Congratulations! Your manuscript is now with our production department. 

Kind regards, 

on behalf of

Dr. Arthur J. Lustig 

Academic Editor

PLOS ONE